# Principles of mRNA targeting via the *Arabidopsis* m⁶A-binding protein ECT2

**Laura Arribas-Hernández[1]\*[†], Sarah Rennie[2†], Tino Köster[3], Carlotta Porcelli[1], Martin Lewinski[3], Dorothee Staiger[3]\*, Robin Andersson[2]\*, Peter Brodersen[1]\***

[1]University of Copenhagen, Copenhagen Plant Science Center, Copenhagen N, Denmark; [2]Department of Biology, University of Copenhagen, Copenhagen, Denmark; [3]University of Bielefeld, Faculty of Biology, RNA Biology and Molecular Physiology, Bielefeld, Germany

**Abstract** Specific recognition of *N6*-methyladenosine (m⁶A) in mRNA by RNA-binding proteins containing a YT521-B homology (YTH) domain is important in eukaryotic gene regulation. The *Arabidopsis* YTH domain protein ECT2 is thought to bind to mRNA at URU(m⁶A)Y sites, yet RR(m⁶A)CH is the canonical m⁶A consensus site in all eukaryotes and ECT2 functions require m⁶A-binding activity. Here, we apply iCLIP (*i*ndividual nucleotide resolution *cr*oss*l*inking and *i*mmuno*p*recipitation) and HyperTRIBE (*t*argets of *R*NA-binding proteins *i*dentified *b*y *e*diting) to define high-quality target sets of ECT2 and analyze the patterns of enriched sequence motifs around ECT2 crosslink sites. Our analyses show that ECT2 does in fact bind to RR(m⁶A)CH. Pyrimidine-rich motifs are enriched around, but not at m⁶A sites, reflecting a preference for *N6*-adenosine methylation of RRACH/GGAU islands in pyrimidine-rich regions. Such motifs, particularly oligo-U and UNUNU upstream of m⁶A sites, are also implicated in ECT2 binding via its intrinsically disordered region (IDR). Finally, URUAY-type motifs are enriched at ECT2 crosslink sites, but their distinct properties suggest function as sites of competition between binding of ECT2 and as yet unidentified RNA-binding proteins. Our study provides coherence between genetic and molecular studies of m⁶A-YTH function in plants and reveals new insight into the mode of RNA recognition by YTH domain-containing proteins.

\*For correspondence:
laura.arribas@bio.ku.dk (LA-H);
dorothee.staiger@uni-bielefeld.
de (DS);
robin@bio.ku.dk (RA);
PBrodersen@bio.ku.dk (PB)

[†]These authors contributed equally to this work

**Competing interest:** The authors declare that no competing interests exist.

## Introduction

*N6*-methyladenosine (m⁶A) is the most abundant modified nucleotide in eukaryotic mRNA bodies. It is required for embryonic development and stem cell differentiation in several animals and plants (*Zhong et al., 2008*; *Batista et al., 2014*; *Ping et al., 2014*; *Geula et al., 2015*; *Zhang et al., 2017*) and for the control of the meiotic program in yeast (*Shah and Clancy, 1992*; *Clancy et al., 2002*; *Agarwala et al., 2012*). Most *N6*-adenosine methylation of mRNA is catalyzed in the nucleus (*Salditt-Georgieff et al., 1976*; *Ke et al., 2017*; *Huang et al., 2019*) by a highly conserved, multimeric methylase (the m⁶A 'writer'; *Balacco and Soller, 2019*) whose catalytic core consists of the heterodimer METTL3/METTL14 (MTA/MTB in plants; *Bokar et al., 1997*; *Zhong et al., 2008*; *Liu et al., 2014*). In addition, a number of highly conserved proteins is required for *N6*-methylation in vivo (*Balacco and Soller, 2019*). The strong conservation of these core factors suggests that the biochemical basis of *N6*-adenosine methylation is common in eukaryotes, and indeed, m⁶A occurs in the consensus site RR(m⁶A̲)CH (R = G/A, H = A/C/U), primarily in 3'-UTRs in vertebrates, plants, and fungi that possess the canonical METTL3/METTL14 methyltransferase (*Bodi et al., 2012*; *Dominissini et al., 2012*; *Meyer et al., 2012*; *Schwartz et al., 2013*; *Luo et al., 2014a*; *Zhao et al., 2017*; *Miao et al., 2020*; *Parker et al., 2020*). Conversely, the characteristic motif and gene body location is not detected in organisms that lack METTL3/METTL14 homologs, such as the nematode *Caenorhabditis elegans* (*Sendinc et al., 2020*) and bacteria (*Deng et al., 2015*).

**eLife digest** Genes are strings of genetic code that contain instructions for producing a cell's proteins. Active genes are copied from DNA into molecules called mRNAs, and mRNA molecules are subsequently translated to create new proteins. However, the number of proteins produced by a cell is not only limited by the number of mRNA molecules produced by copying DNA. Cells use a variety of methods to control the stability of mRNA molecules and their translation efficiency to regulate protein production. One of these methods involves adding a chemical tag, a methyl group, onto mRNA while it is being created. These methyl tags can then be used as docking stations by RNA-binding proteins that help regulate protein translation.

Most eukaryotic species – which include animals, plants and fungi – use the same system to add methyl tags to mRNA molecules. One methyl tag in particular, known as m⁶A, is a well-characterised docking site for a particular type of RNA-binding protein that goes by the name of ECT2 in plants. However, in the flowering plant *Arabidopsis thaliana*, ECT2 was thought to bind to an mRNA sequence different from the one normally carrying the chemical tag, creating obvious confusion about how the system works in plants.

Arribas-Hernández, Rennie et al. investigated this question using advanced large-scale biochemical techniques, and discovered that conventional m⁶A methyl tags are indeed used by ECT2 in *Arabidopsis thaliana*. The confusion likely arose because the sequence ECT2 was thought bind is often located in close proximity to the m⁶A tags, possibly acting as docking stations for proteins that can influence the ability of ECT2 to bind mRNA. Arribas-Hernández, Rennie et al. also uncovered additional mRNA sequences that directly interact with parts of ECT2 previously unknown to participate in mRNA binding.

These findings provide new insights into how chemical labels in mRNA control gene activity. They have broad implications that extend beyond plants into other eukaryotic species, including humans. Since this chemical labelling system has a major role in controlling plant growth, these findings could be leveraged in biotechnology applications to improve crop yields and enhance plant-based food production.

m⁶A may impact mRNA function by different mechanisms, including the creation of binding sites for reader proteins that specifically recognize m⁶A in mRNA (*Dominissini et al., 2012*; *Fu et al., 2014*; *Meyer and Jaffrey, 2014*). The best understood class of readers contains a so-called YT521-B homology (YTH) domain (*Stoilov et al., 2002*) of which two phylogenetic groups, YTHDF and YTHDC, have been defined (*Patil et al., 2018*; *Balacco and Soller, 2019*). The YTH domain harbors a hydrophobic methyl-binding pocket that increases the affinity of m⁶A-containing RNA by more than 10-fold compared to unmethylated RNA (*Li et al., 2014b*; *Luo and Tong, 2014b*; *Theler et al., 2014*; *Xu et al., 2014*; *Zhu et al., 2014*). Apart from interactions with the methylated adenosine and the purine at the −1 position, YTH domain-RNA interactions mostly involve the sugar-phosphate backbone of the RNA (*Luo and Tong, 2014b*; *Theler et al., 2014*; *Xu et al., 2014*). That is consistent with only mild reductions in the binding affinity of the YTH domain of human YTHDC1 upon substitution of nucleotides −2, +1, and +3 that abrogate the canonical RR(m⁶A)CH motif (*Xu et al., 2014*), and poor sequence specificity of RNA binding by isolated YTH domains of human YTHDF1, YTHDF2, and YTHDC1 (*Arguello et al., 2019*). Thus, the methyltransferase complex gives the sequence specificity, while YTH domain proteins may bind to m⁶A-containing RNA regardless of the identity of the immediately adjacent nucleotides.

YTHDF proteins are typically cytoplasmic and consist of a long N-terminal intrinsically disordered region (IDR) followed by the globular YTH domain (*Patil et al., 2018*). Because the affinity of isolated YTH domains for m⁶A-containing RNA is modest, typically with dissociation constants on the order of 0.1–1 µM (*Li et al., 2014b*; *Luo and Tong, 2014b*; *Theler et al., 2014*; *Xu et al., 2014*; *Zhu et al., 2014*), it has been suggested that the IDR may participate in RNA binding (*Patil et al., 2018*). Nonetheless, the clearest evidence for functions of the IDRs in YTHDF proteins reported thus far includes direct interactions with effectors such as the CCR4-NOT complex in mammalian cells (*Du et al., 2016*), and the ability to cause liquid-liquid phase transition when sufficiently high local concentrations are reached (*Arribas-Hernández et al., 2018*; *Gao et al., 2019*; *Ries et al., 2019*; *Fu and Zhuang, 2020*; *Wang et al., 2020*).

The YTHDF family comprises 11 proteins in *Arabidopsis* that are referred to as EVOLUTIONARILY CONSERVED C-TERMINAL REGION1-11 (ECT1-11) (*Li et al., 2014c*; *Scutenaire et al., 2018*). ECT2, ECT3, and ECT4 are expressed in rapidly dividing cells of root, leaf, and flower primordia, and genetic analyses have revealed their general importance in organogenesis (*Arribas-Hernández et al., 2018*; *Arribas-Hernández et al., 2020*). Importantly, the biological functions of ECT2/ECT3/ECT4 described thus far are shared with those of m⁶A writer components and, where tested, have been shown to depend on intact m⁶A-binding pockets, strongly suggesting that the basis for the observed phenotypes in *ect2/ect3/ect4* mutants is defective regulation of m⁶A-modified mRNA targets (*Bodi et al., 2012*; *Shen et al., 2016*; *Ružička et al., 2017*; *Arribas-Hernández et al., 2018*; *Scutenaire et al., 2018*; *Wei et al., 2018*; *Arribas-Hernández et al., 2020*). Despite the progress in identifying biological functions of plant m⁶A-YTHDF axes, a number of fundamental questions regarding their molecular basis remains unanswered. For example, it is unclear whether sequence determinants in addition to m⁶A are important for mRNA target association of ECT proteins in vivo, the mRNA targets of ECT2/ECT3/ECT4 responsible for the developmental delay of *ect2/ect3/(ect4)* mutants have not been identified, and it is not clear what the effects of ECT2/ECT3/ECT4 binding to them may be (*Arribas-Hernández and Brodersen, 2020*). Clearly, robust identification of the mRNA targets directly bound by ECT proteins is key to obtain satisfactory answers to all of these questions. Towards that goal, formaldehyde crosslinking and immunoprecipitation (FA-CLIP) was used to identify mRNA targets of ECT2 (*Wei et al., 2018*). Nonetheless, because formaldehyde, in contrast to UV illumination, generates both protein-protein and protein-RNA crosslinks, it is not an ideal choice for identification of mRNAs bound directly by a protein of interest (see *Arribas-Hernández and Brodersen, 2020* for a discussion). In particular, this problem concerns the unexpected conclusion that ECT2 binds to the 'plant-specific consensus motif' URU(m⁶A)Y (Y = U/C), not RR(m⁶A)CH (*Wei et al., 2018*). Thus, the field of gene regulation via m⁶A-YTHDF modules in plants is in a state of confusion: on the one hand, m⁶A mapping (*Luo et al., 2014a*; *Wan et al., 2015*; *Shen et al., 2016*; *Duan et al., 2017*; *Anderson et al., 2018*; *Miao et al., 2020*; *Parker et al., 2020*) and phenotypes of mutants defective in m⁶A writing (*Bodi et al., 2012*; *Shen et al., 2016*; *Ružička et al., 2017*) or m⁶A binding of ECT2/ECT3/ECT4 (*Arribas-Hernández et al., 2018*; *Arribas-Hernández et al., 2020*) suggest that these YTHDF proteins should act via recognition of m⁶A in the RRACH context. On the other hand, the only attempt at a mechanistic understanding of ECT2 function via mRNA target identification concluded that ECT2 binds to a sequence element different from RRACH (*Wei et al., 2018*). To complicate matters further, a number of motifs including not only URUAY, but also UGUAMM (M = A/C), UGWAMH (W = A/U), UGUAWA, and GGAU have been reported to be enriched around m⁶A sites in *Arabidopsis* and other plant species (*Li et al., 2014a*; *Anderson et al., 2018*; *Miao et al., 2020*; *Zhang et al., 2019*; *Zhou et al., 2019*), but it remains unclear whether the adenosines in such motifs are methylated in vivo. Alternatively, these sequence contexts may play a role in guiding m⁶A deposition or ECT recognition nearby, either directly by ECT interaction or indirectly via additional RNA-binding proteins assisting or competing with ECT binding.

To clarify principles underlying mRNA recognition by ECT2, we undertook rigorous analysis of its mRNA-binding sites using two orthogonal methods, the proximity-labeling method HyperTRIBE (*t*argets of *R*NA-binding proteins *i*dentified *by* editing) (*McMahon Aoife et al., 2016*; *Xu et al., 2018*) and iCLIP (*i*ndividual nucleotide resolution *c*ross*l*inking and *i*mmuno*p*recipitation) (*König et al., 2010*). This resulted in identification of high-quality target sets as judged by mutual overlaps and by overlaps with previously reported m⁶A maps from plants at a similar developmental stage (*Shen et al., 2016*; *Parker et al., 2020*). Relying on this high-quality target set, we used the position information inherent to iCLIP and a single-nucleotide resolution m⁶A dataset (*Parker et al., 2020*) to establish six properties of m⁶A-containing mRNA and mRNA targeting by ECT2. (1) RRACH and its variant DRACH (D = R/U) are unequivocally the most highly enriched motifs at m⁶A sites in *Arabidopsis*. (2) ECT2 binds to m⁶A sites in the canonical RRACH context as ECT2 crosslinking sites are preferentially found immediately 5′ to m⁶A sites, and RRACH is enriched immediately 3′ to ECT2 crosslinking sites. (3) GGAU is a minor m⁶A consensus site in plants. (4) U- and U/C-rich motifs are enriched around, but not at, m⁶A sites, and, together with RRACH and GGAU, constitute core elements that distinguish m⁶A-containing 3′-UTRs from non-m⁶A-containing 3′-UTRs in plants. (5) The IDR of ECT2 participates in RNA binding as it crosslinks to target mRNAs at U-rich elements highly abundant upstream of m⁶A sites. (6) Although URUAY, URURU, and similar motifs may crosslink to ECT2, their presence in

m⁶A-containing mRNA disfavors ECT2 binding, consistent with those motifs acting predominantly as sites of interaction for RNA-binding proteins that may compete with ECT2.

## Results

### ADARcd fusions to ECT2 are functional in vivo

HyperTRIBE uses fusion of RNA-binding proteins to the hyperactive E488Q mutant of the catalytic domain of the *Drosophila melanogaster* adenosine deaminase acting on RNA (*Dm*ADAR^E488Qcd) (*Kuttan and Bass, 2012*) to achieve proximity labeling in vivo (*McMahon Aoife et al., 2016*; *Xu et al., 2018*). Targets are identified as those mRNAs that contain adenosine-inosine sites significantly more highly edited than background controls, measured as A-G changes upon reverse transcription and sequencing. To develop material suitable for ECT2 HyperTRIBE, we expressed *AtECT2pro:AtECT2-FLAG-DmADAR^E488Qcd-AtECT2ter'* (henceforth '*ECT2-FLAG-ADAR*') in the single *ect2-1* and triple *ect2-1/ect3-1/ect4-2* (*te234*) knockout backgrounds (*Arribas-Hernández et al., 2018*; *Arribas-Hernández et al., 2020*). We identified lines exhibiting nearly complete rescue of *te234* mutant seedling phenotypes, indicating that the fusion protein was functional (*Figure 1A*). We then used the expression level in complementing lines as a criterion to select lines in the *ect2-1* single mutant background, for which no easily scorable phenotype has been described (*Figure 1—figure supplement 1A*). Lines expressing free *Dm*ADAR^E488Qcd under the control of the endogenous *ECT2* promoter (*AtECT2pro:FLAG-DmADAR^E488Qcd-AtECT2ter*; henceforth *FLAG-ADAR*) at levels similar to or higher than those of the fusion lines (*Figure 1—figure supplement 1A,B*) were used to control for background editing after verification that *FLAG-ADAR* expression did not result in phenotypic abnormalities in Col-0 WT plants (*Figure 1A*).

### The ECT2-ADARcd fusion imparts adenosine-to-inosine editing of target mRNAs in planta

To identify ECT2 HyperTRIBE targets (HT-targets), we sequenced mRNA from dissected root tips and shoot apices of 10-day-old seedlings of *ect2-1/ECT2-FLAG-ADAR* and *FLAG-ADAR* transgenic lines using five independent lines of each type as biological replicates to prevent line-specific artifacts. Next, we generated nucleotide base counts for all positions with at least one mismatch across the full set of samples of mapped reads (*Figure 1B*), resulting in a raw list of potential editing positions. This revealed that the amount of editing was clearly higher in the lines expressing the ECT2-FLAG-ADAR fusion protein than in the negative control lines (*Figure 1C*, *Figure 1—figure supplement 1C*). To identify positions with significantly higher editing rates in ECT2-FLAG-ADAR lines compared to controls, we developed a new approach to detect differential editing (*Figure 1B*) described in detail by *Rennie et al., 2021*. Briefly, the hyperTRIBE**R** method of detecting differential editing exploits the powerful statistical capabilities of a method originally designed to detect differential exon usage (*Anders et al., 2012*). It efficiently takes replicates and possible differences in expression into account, resulting in high power to detect sites despite the generally low editing proportions that we found in our data (*Figure 1D*). As expected, the tendency towards higher editing proportions in fusion lines compared to controls was even more pronounced after filtering nonsignificantly edited sites (*Figure 1C*, *Figure 1—figure supplement 1C*). Three additional properties of the resulting editing sites indicate that they are the result of ADARcd activity guided by its fusion to ECT2. First, the vast majority of significant hits corresponded to A-to-G transitions (*Figure 1—figure supplement 1D*). Second, the consensus motif at the edited sites matched the sequence preference of *Dm*ADAR^E488Qcd (5′ and 3′ nearest-neighbor preference of U>A>C>G and G>C>A~U, respectively [*Xu et al., 2018*; *Figure 1E*, *Figure 1—figure supplement 1E*]), with highly edited sites more closely matching the optimal sequence context than lowly edited ones (*Figure 1—figure supplement 1F*). Third, principal component analysis of editing proportions at significant sites over the different lines clearly separated the ECT2-FLAG-ADAR fusion lines from the control lines (*Figure 1F*, *Figure 1—figure supplement 1G*). Application of subsequent filtering steps, including removal of non-(A-to-G) mismatches and of potential line-specific single-nucleotide variants (see Materials and methods), resulted in a final list of 16,176 edited sites for aerial tissues and 19,242 for roots, corresponding to 4864 and 5052 genes (ECT2 HT-targets), respectively (*Figure 1B*, *Supplementary file 1*). In both cases, this represents 27% of all expressed genes. We note that the editing proportions were generally low (*Figure 1D*) compared

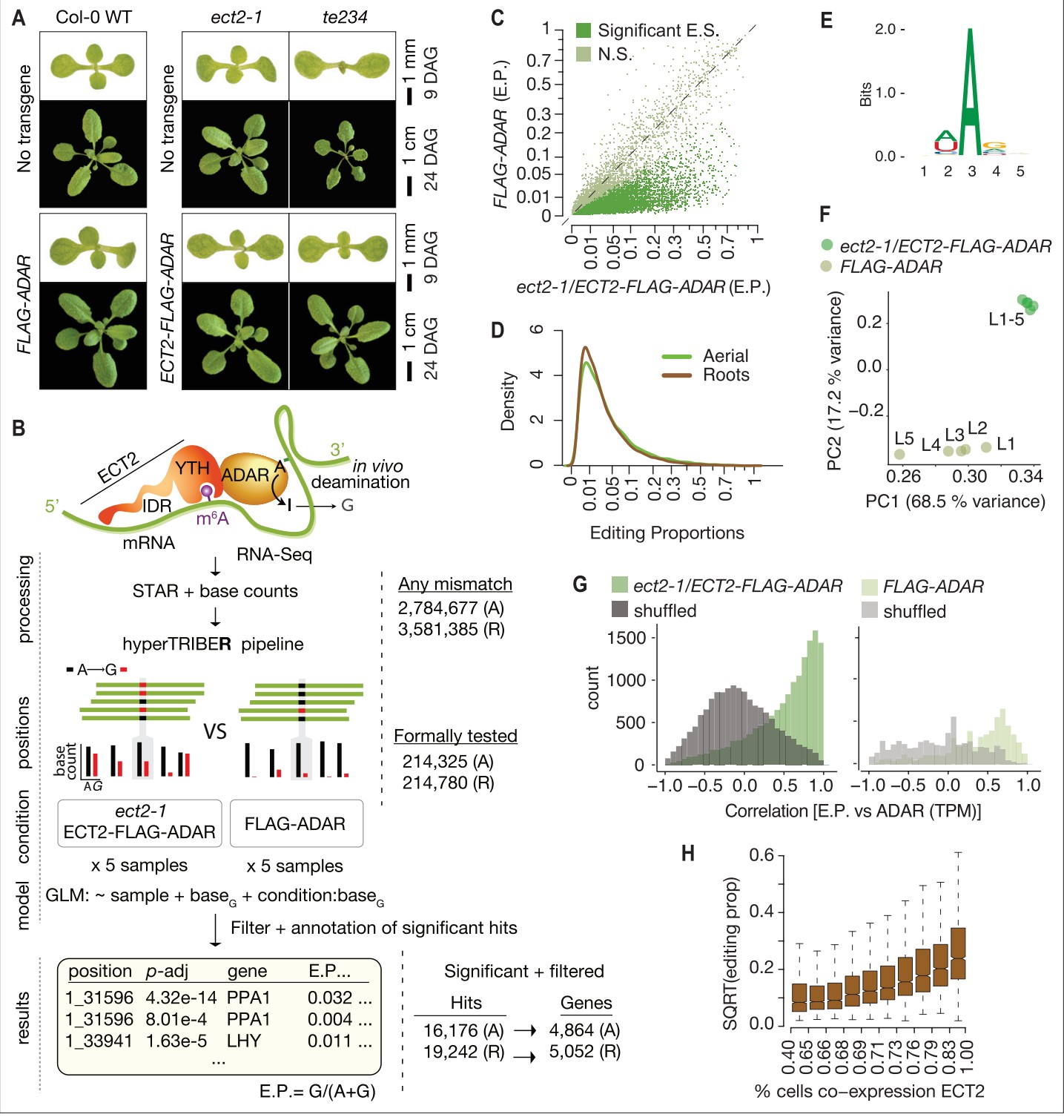

**Figure 1.** *Drosophila* ADARcd fused to ECT2 can edit target mRNAs in vivo in plants. (**A**) Phenotypes of wild type, *ect2-1* and *te234* mutants with (lower panels) or without (upper panels) *ECT2-FLAG-ADAR* or *FLAG-ADAR* transgenes, at 9 or 24 days after germination (DAG). (**B**) Experimental design for ECT2-HyperTRIBE (ECT2-HT) target identification and hyperTRIBE**R** pipeline (*Rennie et al., 2021*). Nucleotide base counts quantified from mapped RNA-seq libraries were passed into the hyperTRIBE**R** pipeline to call significant editing sites, which were further filtered and annotated. The number of sites in either aerial (A, dissected apices) or root (R, root tips) tissues considered at each stage of the analysis is indicated. GLM, generalized linear model; E.P., editing proportion. (**C**) Scatterplot of the editing proportions of potential and significant editing sites (E.S.) in aerial tissues of *ect2-1/ECT2-FLAG-ADAR* lines compared to the *FLAG-ADAR* controls. Significant sites are highlighted in vivid green. N.S., not significant. (**D**) Density of editing

*Figure 1 continued on next page*

*Figure 1 continued*

proportions for significant editing sites in aerial tissues and roots of *ect2-1/ECT2-FLAG-ADAR* lines. (**E**) Consensus motif identified at significant editing sites in aerial tissues of *ect2-1/ECT2-FLAG-ADAR* lines. (**F**) Principal component analysis of editing proportions at significant editing sites in samples with aerial tissues. (**G**) Distribution of the correlations between editing proportions and ADAR expression (TPM) for significant editing sites in aerial tissues of either *ect2-1/ECT2-FLAG-ADAR* or *FLAG-ADAR* lines. Background correlations (gray) are based on randomly shuffling ADAR expression for each site. (**H**) Boxplots showing the mean editing proportions as a function of the proportion of cells co-expressing *ECT2*, calculated based on single cell RNA-seq in roots (*Denyer et al., 2019*). For panels **C**, **E**, **F**, and **G**, comparable analyses in both aerial and root tissues are shown in the *Figure 1—figure supplement 1*.

The online version of this article includes the following figure supplement(s) for figure 1:

**Figure supplement 1.** *Drosophila* ADARcd fused to ECT2 can edit target mRNAs in vivo in plants (extended data, aerial and root tissues).

**Figure supplement 1—source data 1.** Uncropped labeled panels and raw image files: *Figure 1—figure supplement 1A*.

to previous work in *Drosophila* (*Xu et al., 2018*), perhaps in part due to the limited number of cells that express ECT2 (*Arribas-Hernández et al., 2018*; *Arribas-Hernández et al., 2020*). Indeed, the *ADAR* expression level (TPMs) correlated strongly with editing proportions among *ECT2-FLAG-ADAR* lines (*Figure 1G*, *Figure 1—figure supplement 1H*), and editing proportions were higher for target mRNAs that are coexpressed with *ECT2* in a large percentage of cells according to single-cell RNA-seq (*Denyer et al., 2019*; *Figure 1H*), lending further support to the conclusion that the observed editing is ADAR-specific and driven to target mRNAs by ECT2. Hence, HyperTRIBE can be used to identify targets of RNA-binding proteins in planta.

## HyperTRIBE is highly sensitive and identifies primarily m⁶A-containing transcripts as ECT2 targets

To evaluate the properties of ECT2 HT-targets, we first noted that most of them were common between root and aerial tissues (*Figure 2A*), as expected given the recurrent function of ECT2 in stimulating cell division in all organ primordia (*Arribas-Hernández et al., 2020*). In agreement with this result, most of the targets specific to root or aerial tissues were simply preferentially expressed in either tissue (*Figure 2B*). Moreover, the significant editing sites in roots and aerial tissues had a considerable overlap (*Figure 2A*), and their editing proportions were similar in the two tissues (*Figure 2C*).

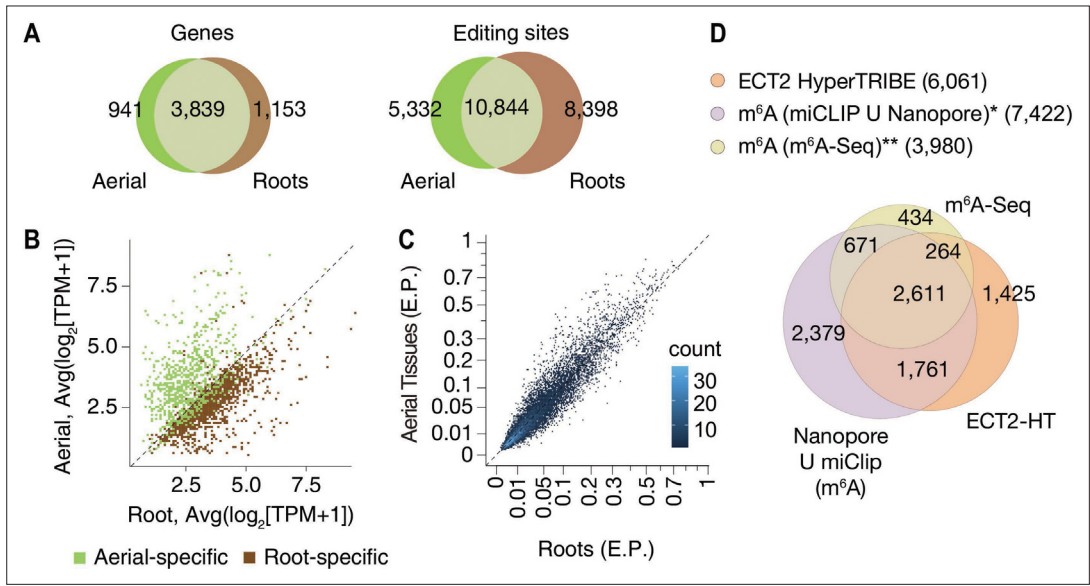

**Figure 2.** HyperTRIBE identifies m⁶A-reader targets in plants. (**A**) Overlap between ECT2-HT targets (genes and editing sites) in roots and aerial tissues, based on genes commonly expressed in both tissues. (**B**) Scatterplot showing the expression levels in roots and aerial tissues (mean log₂(TPM+1) over the five ECT2-HT control samples) of the genes identified as aerial or root-specific targets. (**C**) Scatterplot of the editing proportions (E.P.) of significant editing sites in ECT2-HT for aerial vs root tissues. (**D**) Overlap between ECT2-HT targets and m⁶A-containing genes. *\*Parker et al., 2020*; \*\* *Shen et al., 2016*.

Of most importance, we observed a large overlap between the ECT2 HT-targets and m⁶A-containing transcripts mapped by different methods in seedlings (*Shen et al., 2016*; *Parker et al., 2020*) as more than 76% of ECT2 HT-targets had m⁶A support by either study (*Figure 2D*). These results validate our HyperTRIBE experimental setup and data analysis, and confirm that ECT2 binds predominantly to m⁶A-containing transcripts in vivo.

## ECT2-mCherry can be specifically UV-crosslinked to target RNA in vivo

We next moved on to independent target and binding site identification via iCLIP (*Figure 3A*). We used transgenic lines expressing functional ECT2-mCherry under the control of the endogenous *ECT2* promoter in the *ect2-1* knockout background (*Arribas-Hernández et al., 2018*; *Arribas-Hernández et al., 2020*) to co-purify mRNAs crosslinked to ECT2 for iCLIP. Lines expressing the *ECT2^W464A*-*mCherry* variant were used as negative controls because this Trp-to-Ala mutation in the hydrophobic methyl-binding pocket of the YTH domain abrogates the increased affinity for m⁶A-RNA (*Li et al., 2014b*; *Xu et al., 2014*; *Zhu et al., 2014*). Accordingly, the point mutant behaves like a null allele in plants despite its wild-type-like expression pattern and level (*Arribas-Hernández et al., 2018*; *Arribas-Hernández et al., 2020*).

To test the feasibility of iCLIP, we first assessed the specificity of RNA co-purified with ECT2-mCherry after UV illumination of whole seedlings by 5'-radiolabeling of the immunoprecipitated RNP complexes followed by SDS-PAGE. These tests showed that substantially more RNA co-purifies with wild-type ECT2 than with ECT2^W464A upon UV-crosslinking, and that no RNA is detected without UV irradiation or from irradiated plants of non-transgenic backgrounds (*Figure 3B*, *Figure 3—figure supplement 1A*). RNAse and DNAse treatments also established that the co-purified nucleic acid is RNA (*Figure 3—figure supplement 1B*). Thus, UV crosslinking of intact *Arabidopsis* seedlings followed by immunopurification successfully captures ECT2-RNA complexes that exist in vivo. Curiously, although the pattern of ECT2-RNA complexes with bands migrating at ~110 and 55kDa is highly reproducible, it does not correspond to the majority of the purified ECT2-mCherry protein, which runs at ~125kDa in SDS-PAGE (*Figure 3B and C*). A variety of control experiments (*Figure 3—figure supplement 1C-E*), most importantly the disappearance of additional bands with use of an N-terminal rather than a C-terminal tag (*Figure 3C and D*), indicate that the band pattern arises as a consequence of proteolytic cleavage of the N-terminal IDR in the lysis buffer, such that fragments purified using the C-terminal mCherry tag include the YTH domain with portions of the IDR of variable lengths (*Figure 3—figure supplement 2*). Comparative analysis of RNA in 55-kDa and 110–125-kDa complexes may, therefore, provide insight into the possible role of the N-terminal IDR of ECT2 in mRNA binding (*Figure 3E*), an idea consistent with the comparatively low polynucleotide kinase labeling efficiency of full-length ECT2-mCherry-mRNA complexes (~125kDa) (*Figure 3B*, *Figure 3—figure supplement 2*). Thus, we prepared separate iCLIP libraries from RNA crosslinked to ECT2-mCherry/ECT2^W464A-mCherry that migrates at ~110–280kDa ('110-kDa band') and at ~55–75kDa ('55-kDa band') (*Figure 3—figure supplement 3*) to investigate the possible existence of IDR-dependent crosslink sites, and thereby gain deeper insights into the mode of YTHDF binding to mRNA in vivo.

## ECT2-mCherry iCLIP peaks are enriched in the 3'-UTR of mRNAs

We identified a total of 15,960 iCLIP 'peaks' or crosslink sites (i.e., single-nucleotide positions called by PureCLIP from mapped iCLIP reads [*Krakau et al., 2017*]) in 2281 genes from the 110-kDa band of wild-type ECT2-mCherry (henceforth referred to as ECT2 iCLIP peaks and targets, respectively). In the corresponding 55-kDa band, 4549 crosslink sites in 1127 genes were called, 93% of them contained in the 110-kDa target set (*Figure 3F and G*, *Figure 3—figure supplement 4*, *Supplementary file 2*). We note that these numbers perfectly agree with the idea of the 55-kDa band containing only YTH domain crosslink sites, while the full length may also include IDR crosslink sites. Importantly, for both libraries, the majority of crosslink sites mapped to the 3'-UTRs of mRNAs (*Figure 3H*, see *Figure 4A*, and *Figure 4—figure supplement 1* for more examples), coincident with the main location of m⁶A (*Figure 4B*; *Parker et al., 2020*). Accordingly, the 3'-UTR specificity was largely lost in RNA isolated from 55-kDa ECT2^W464A (*Figure 3H*), for which neither YTH domain nor IDR binding to RNA can be expected. Finally, iCLIP targets in full-length (110-kDa band) ECT2 WT and ECT2^W464A overlapped only marginally (*Figure 3G*), providing molecular proof of the dependence of m⁶A-binding activity for ECT2 function. Nonetheless, the bias towards occurrence in the 3'-UTR was only reduced, not abolished, for

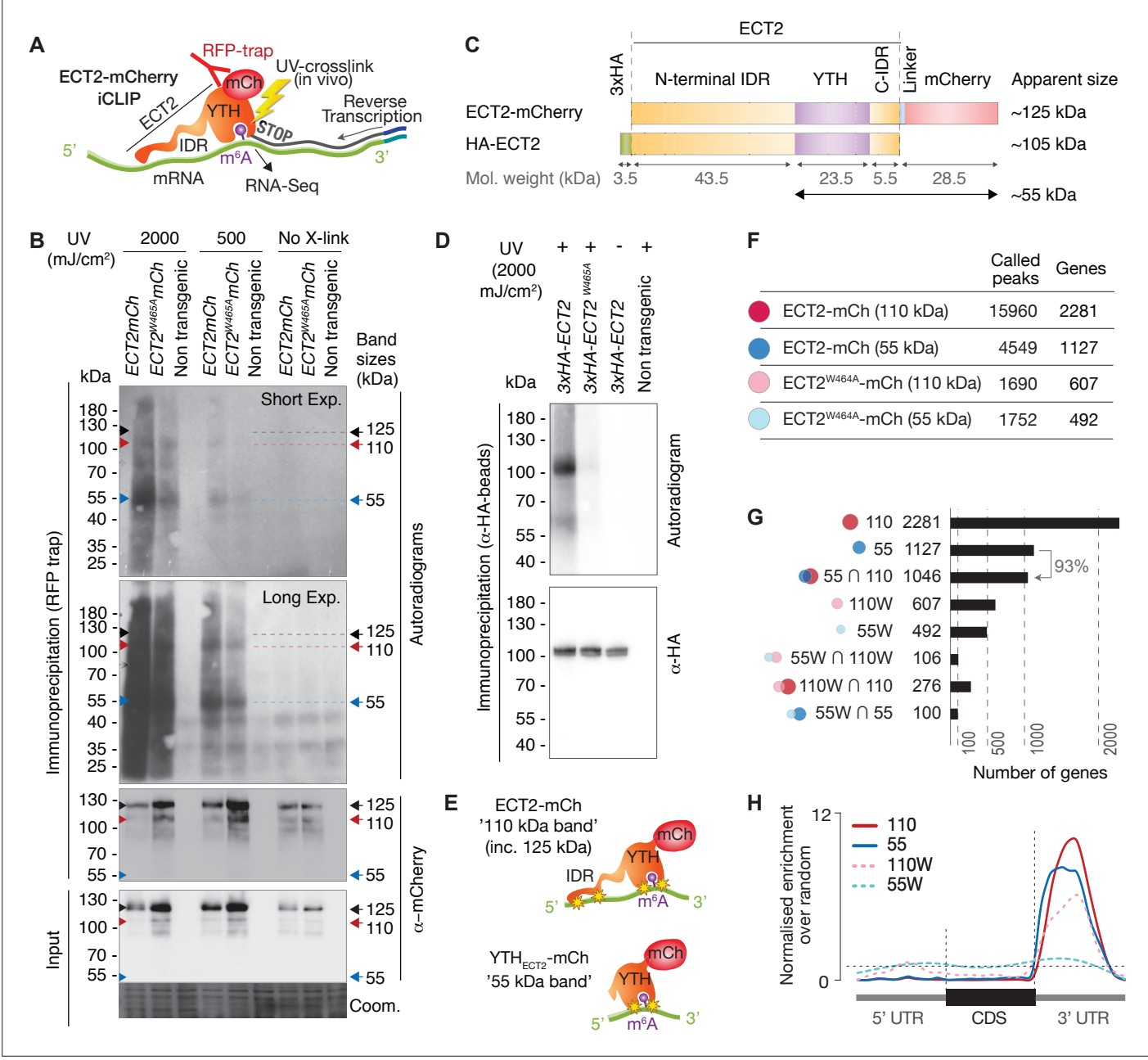

**Figure 3.** RNA-binding properties of ECT2 revealed by CLIP. (**A**) iCLIP experimental design. (**B**) Upper panels: autoradiogram (top) and α-mCherry protein blot (below) of RFP-trap immuno-purifications. Samples are cell extracts from 12-day-old seedlings expressing *ECT2-mCherry* or *ECT2^{W464A}-mCherry* in the *ect2-1* mutant background after in vivo UV-crosslinking as indicated, and subjected to DNase digestion, partial RNase digestion, and 5'-$^{32}$P labeling of RNA. Non-transgenic, Col-0 wild type. Lower panels: α-mCherry protein blot of the same extracts before immunoprecipitation (input) and Coomassie staining of the membrane. Sizes corresponding to full length ECT2-mCherry (~125 kDa) and the most apparent RNA bands are indicated with arrows. A repeat of the experiment with independently grown and crosslinked tissue is shown in the *Figure 3—figure supplement 1A*. (**C**) Schematic representation of ECT2-mCherry and HA-ECT2 fusion proteins with their apparent size (electrophoretic mobility). The molecular weight of each region is indicated. Notice that IDRs tend to show higher apparent sizes (lower electrophoretic mobility) than globular domains. (**D**) Equivalent to **B** with lines expressing *3xHA-ECT2* variants in the *ect2-1* background, α-HA immuno-purifications and α-HA detection by western blot. (**E**) Cartoon illustrating the nature of the bands of labelled RNA co-purifying with ECT2-mCherry. Yellow stars indicate possible crosslinking sites. (**F**) Number of called peaks and genes detected from the four iCLIP libraries sequenced for this study (*Figure 3—figure supplement 3*). (**G**) Upset plot showing single and pairwise combinations of genes for the four sequenced iCLIP libraries. Additional intersections can be found in the *Figure 3—figure supplement 4*. (**H**) Metagene profiles depicting the enrichment along the gene body (5'UTR, CDS or 3'UTR) of the called iCLIP peaks detailed in **F**.

The online version of this article includes the following source data and figure supplement(s) for figure 3:

*Figure 3 continued on next page*

*Figure 3 continued*

**Source data 1.** Uncropped labelled panels and raw image files - *Figure 3B, D*.

**Figure supplement 1.** UV-crosslinked RNA co-purifies with ECT2-mCherry in a pattern that depends on the proteolytic cleavage of the ECT2 intrinsically disordered region (IDR) in the lysate.

**Figure supplement 1—source data 1.** Uncropped labeled panels and raw image files: *Figure 3—figure supplement 1A-E*.

**Figure supplement 2.** Illustration of RNA-binding properties of ECT2 revealed by CLIP.

**Figure supplement 3.** iCLIP Libraries.

**Figure supplement 3—source data 1.** Uncropped labeled panels and raw image files: *Figure 3—figure supplement 3A-C*.

**Figure supplement 4.** Analysis of ECT2 iCLIP Libraries.

crosslinks to the full-length ECT2$^{W464A}$ protein, providing another indication that the IDR itself is able to associate with RNA-elements in 3′-UTRs (*Figure 3H*). We elaborate further on this important point by analysis of IDR-specific crosslinks to wild-type ECT2 after in-depth validation of sets of ECT2 target mRNAs and determination of the sequence motifs enriched around m⁶A and ECT2 crosslink sites.

## iCLIP sites tend to be in the vicinity of HyperTRIBE editing sites

To evaluate the congruence of the results obtained by iCLIP and HyperTRIBE, we investigated the cumulative number of iCLIP sites as a function of distance to the nearest editing site determined by HyperTRIBE. This analysis showed a clear tendency for iCLIP peaks called with ECT2$^{WT}$-mCherry, but not for ECT2$^{W464A}$-mCherry, to be in the vicinity of editing sites (*Figure 4C*), indicating that the majority of called iCLIP peaks identify genuine ECT2-binding sites on mRNAs. Similar tendencies of proximity between iCLIP peaks and HyperTRIBE editing sites were previously observed for a *Drosophila* hnRNP protein (*Xu et al., 2018*). Although manual inspection of individual target genes confirmed these tendencies, it also revealed that ADAR-edited sites are too dispersed around iCLIP peaks to give precise information on the actual ECT2-binding sites (*Figure 4A*, *Figure 4—figure supplement 1*). Therefore, we used both HyperTRIBE and iCLIP for gene target identification, but relied on iCLIP peaks for motif analyses.

## ECT2 targets identified by iCLIP and HyperTRIBE overlap m⁶A-containing transcripts

To examine the quality of our target identification in further detail, we analyzed the overlap between ECT2 targets identified by iCLIP and HyperTRIBE. This analysis also included m⁶A mapping data obtained with either m⁶A-seq (*Shen et al., 2016*) or the single-nucleotide resolution methods miCLIP and Nanopore sequencing (*Parker et al., 2020*) as young seedlings were used in all cases. ECT2 targets identified by iCLIP and HyperTRIBE showed clear overlaps, both with each other and with m⁶A-containing transcripts, further supporting the robustness of ECT2 target identification via combined iCLIP and HyperTRIBE approaches (*Figure 4D*, upper panel, *Figure 4—figure supplement 2*). Importantly, although some m⁶A targets are expected not to be bound by ECT2 because of the presence of MTA in cells that do not express ECT2 (*Arribas-Hernández et al., 2020*), only 18% of the high-confident set of m⁶A-containing genes (with support from miCLIP and Nanopore) did not overlap with either ECT2 iCLIP or HT target sets (*Figure 4—figure supplement 2*, arrow). We also observed that HyperTRIBE identifies approximately three times more ECT2 targets than iCLIP, possibly because of the bias towards high abundance inherent to purification-based methods like iCLIP (*Wheeler et al., 2018*). To test this idea, we compared the distribution of target mRNAs identified by the different techniques across nine expression bins. As expected, a bias towards highly abundant transcripts was evident for iCLIP-identified targets compared to HyperTRIBE (*Figure 4E*). We also observed a similar bias for m⁶A-containing transcripts detected by miCLIP, another purification-based method, and in the Nanopore dataset (*Figure 4E*), probably explained by its relatively low sequencing depth (*Parker et al., 2020*). These observations also suggest that the higher sensitivity of HyperTRIBE (analyzed in detail in *Figure 4—figure supplement 3*) explains the lack of m⁶A support (by Nanopore or miCLIP) for 28% of ECT2 HT-targets (1689) compared to only 4% (83) of ECT2 iCLIP targets (*Figure 4D*, *Figure 4—figure supplement 2*, upper row) since HT-targets may simply include genes that escape detection by m⁶A mapping methods due to low expression. Indeed, ECT2-HT targets without any m⁶A support were distributed in lower-expression bins compared to those with m⁶A support (*Figure 4F*).

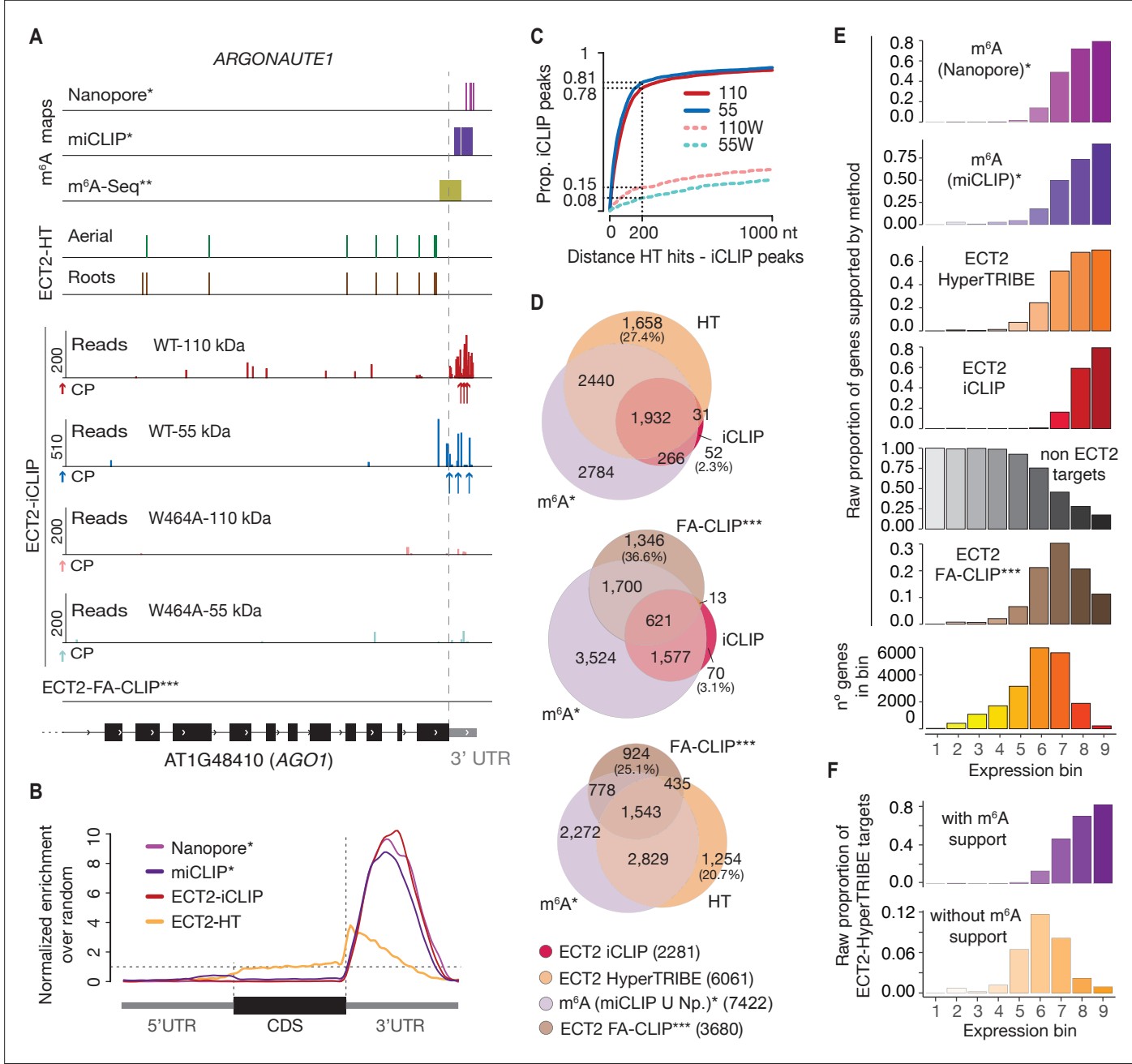

**Figure 4.** CLIP identifies bona-fide ECT2 targets. (**A**) Example of an ECT2 target (*AGO1*) showing the distribution of m⁶A sites*, **, ECT2-iCLIP reads and peaks, ECT2-HT edited sites, and FA-CLIP peaks*** along the transcript. CP, called peaks. See more examples in the *Figure 4—figure supplement 1*. (**B**) Metagene profiles comparing the distributions along the gene body of ECT2-mCherry iCLIP peaks (wild type, 110-kDa band), ECT2-HT editing sites (in roots and aerial tissues) and m⁶A sites*. (**C**) Proportion of ECT2 iCLIP peaks within a given distance from the nearest ECT2-HT edited site. Numbers indicated on the y-axis show the proportion of ECT2 iCLIP peaks less than or equal to 200 nt from the nearest ECT2-HT edited site. (**D**) Overlap between genes supported as containing m⁶A or ECT2 targets by the different techniques indicated. The ECT2-HT target set includes the sum of targets identified in root and aerial tissues. Additional overlaps are shown in the *Figure 4—figure supplement 2*. (**E**) Proportions of genes in each expression bin either containing m⁶A or supported as ECT2 targets by the indicated techniques. (**F**) Proportion of ECT2-HT targets with or without support from m⁶A data (Nanopore*, miCLIP* or m⁶A-Seq**) in each expression bin. * *Parker et al., 2020*; ** *Shen et al., 2016*; *** *Wei et al., 2018*.

The online version of this article includes the following figure supplement(s) for figure 4:

**Figure supplement 1.** Distribution of m⁶A and ECT2 sites on ECT2 targets.

**Figure supplement 2.** Overlaps between m⁶A-containing genes and ECT2 targets datasets.

**Figure supplement 3.** Characteristics of ECT2-HyperTRIBE editing sites relative to target expression levels.

Intriguingly, ECT2 FA-CLIP targets (*Wei et al., 2018*) did not show a bias towards highly expressed genes as their distribution over expression bins largely reflected that of the total number of genes (*Figure 4E*), and as many as 37% of FA-CLIP targets did not have m⁶A support (*Figure 4D*, *Figure 4— figure supplement 2*, upper row). In summary, these analyses show that ECT2 iCLIP and HT target sets are in excellent agreement with each other and with independently generated m⁶A maps, and that HyperTRIBE identifies targets below the detection limit of other techniques.

## ECT2 crosslink sites coincide with m⁶A miCLIP sites and are immediately upstream of Nanopore m⁶A sites

To characterize the sequence composition and exact positions of ECT2-binding sites relative to m⁶A, we first used the high resolution of iCLIP data to examine the position of ECT2 crosslink sites relative to m⁶A sites, determined at single-nucleotide resolution (*Parker et al., 2020*). This analysis showed that ECT2 crosslinks in the immediate vicinity, but preferentially upstream (~11 nt) of Nanopore-determined m⁶A sites, with a mild depletion at the exact m⁶A site (*Figure 5A*, upper panel). Furthermore, while m⁶A-miCLIP sites corresponded to m⁶A-Nanopore sites overall, a subset of m⁶A-miCLIP sites were located upstream of m⁶A-Nanopore sites and coincided well with ECT2-iCLIP peaks (*Figure 5A*). This pattern is probably explained by the fact that the UV illumination used in both iCLIP and miCLIP preferentially generates RNA-protein crosslinks involving uridine (*Hafner et al., 2021*), also detectable in the datasets analyzed here (*Figure 5B and C*). Thus, the depletion of ECT2-iCLIP sites at Nanopore-, but enrichment at miCLIP-determined m⁶A sites (*Figure 5A*), might be explained by the absence of uridine within the RR<u>A</u>C core of the m⁶A consensus motif, and perhaps also to some extent by reduced photoreactivity of the m⁶A base stacking with indole side chains of the YTH domain. Furthermore, the fact that nucleotides at −2, +1, and +2 positions are only expected to contribute sugar-phosphate backbone interactions with the YTH domain (*Luo and Tong, 2014b*; *Theler et al., 2014*; *Xu et al., 2014*) may also contribute to the absence of direct crosslinks at the m⁶A site relative to the adjacent bases.

## DRACH, GGAU, and U/Y-rich motifs are the most enriched around m⁶A/ ECT2 sites

The 5′ shift observed for iCLIP and miCLIP sites relative to Nanopore sites might be explained by a higher occurrence of uridines upstream of m⁶A sites, a particularly interesting possibility given the numerous reports of U-rich motifs enriched around m⁶A sites in plants (*Li et al., 2014a*; *Anderson et al., 2018*; *Miao et al., 2020*; *Zhang et al., 2019*; *Zhou et al., 2019*; *Luo et al., 2020*) and animals (*Patil et al., 2016*). To investigate the sequence composition around m⁶A and ECT2 sites, we first performed exhaustive unbiased de novo motif searches using Homer (*Heinz et al., 2010*; *Figure 5— figure supplement 1*) and extracted all candidate motifs, including the m⁶A consensus motif RRACH, as well as GGAU (*Anderson et al., 2018*), URUAY (*Wei et al., 2018*), and several other U-rich sequences. Combined with manually derived candidate motifs (*Figure 5—figure supplement 1B*), we then calculated position weight matrices (PWMs) for a final set of 48 motifs and scanned for their occurrences genome-wide using FIMO (*Grant et al., 2011*; *Figure 5—figure supplements 1 and 2*). This allowed us to determine three key properties. First, the global enrichment of the motifs at locations across the gene body. Second, the total count of occurrences of each motif at m⁶A sites and ECT2-iCLIP crosslink sites compared to a set of sites in non-target mRNAs matching the location within gene bodies of m⁶A/ECT2-iCLIP sites (expected background). Third, the distribution of the motifs relative to m⁶A and ECT2 iCLIP sites. The results of this systematic analysis (*Supplementary file 3*) were used to select those motifs with a more prominent enrichment at or around m⁶A and ECT2 sites (*Figure 5D*). This approach defined two major categories of motifs of outstanding interest, RRACH-like and GGAU on the one side, and a variety of U/Y-rich motifs on the other. *Figure 5D* shows a minimal selection of such motifs, while a more comprehensive compilation is displayed in *Figure 5—figure supplements 3 and 4*. Not surprisingly, RRACH-like motifs were the most highly enriched at m⁶A sites and showed a clear enrichment immediately downstream of ECT2 crosslink sites in our analyses, with the degenerate variant DRACH being the most frequently observed (*Figure 5D*, *Figure 5—figure supplement 3*). Motifs containing GGAU behaved similarly to DRACH, with a sharp enrichment exactly at m⁶A sites and mild enrichment downstream of ECT2 peaks (*Figure 5D*), supporting a previous suggestion of

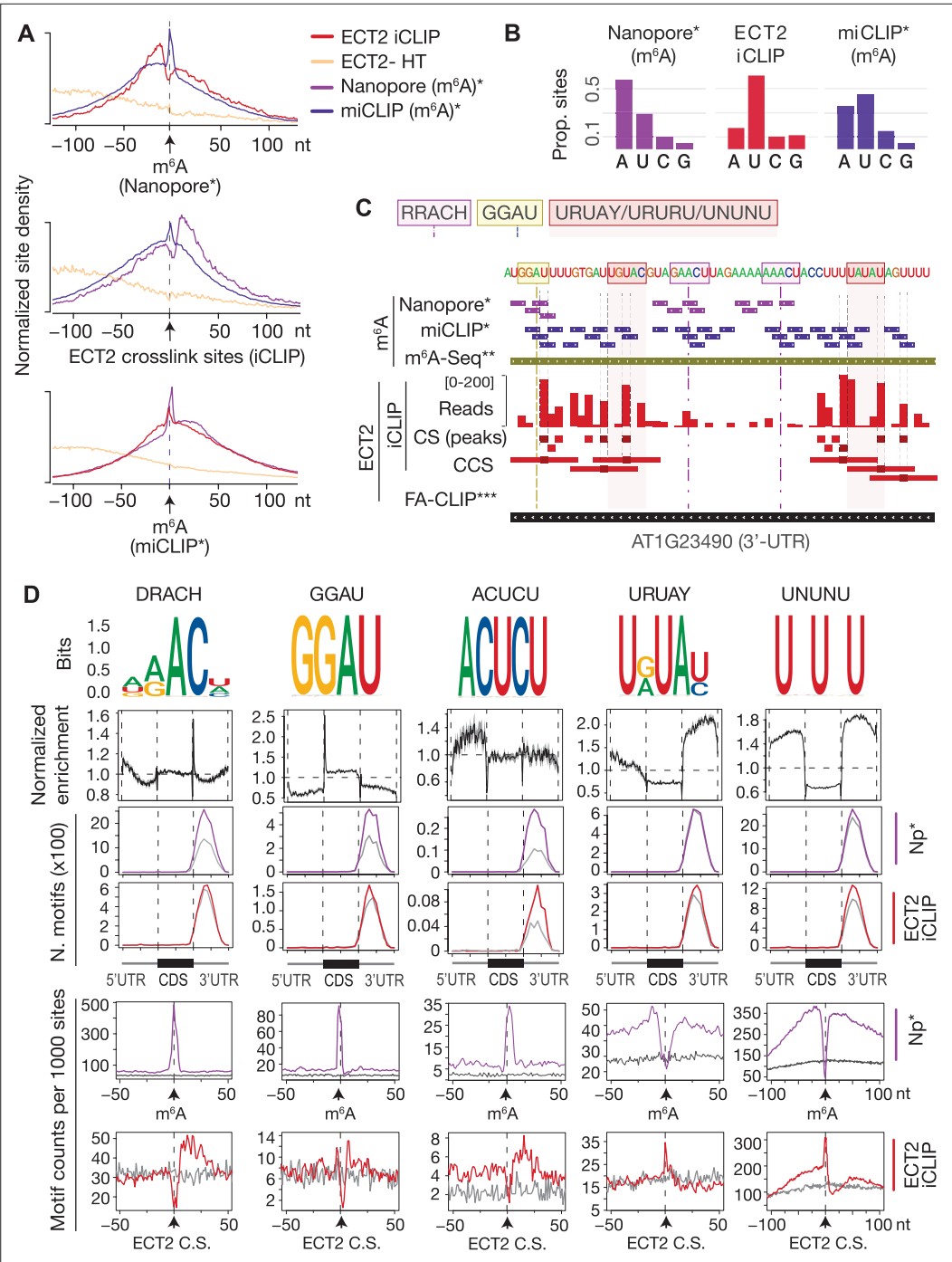

**Figure 5.** ECT2 UV-crosslinks to uridines in the immediate vicinity of DR(m⁶A)CH or GG(m⁶A)U sites. (**A**) Normalized density of sites at and up to +/-100 nt of either m⁶A-Nanopore*, m⁶A-miCLIP* or ECT2-iCLIP sites. (**B**) Proportion of m⁶A and ECT2-iCLIP sites at each nucleotide by the different methods. (**C**) View from IGV browser illustrating the presence of RRACH, GGAU and U-rich motifs in the vicinity of m⁶A and ECT2 sites in the 3'-UTR of AT1G23490 (*ARF1*). CS, crosslink sites; CSS, collapsed crosslink sites. (**D**) Key motifs analyzed in this study. From top to bottom: (1) motif logos for derived position weight matrices (PWMs); (2) normalized enrichment of motif locations across gene body; (3-4) total number of the relevant motif found at m⁶A-Nanopore* (3) or ECT2-iCLIP (4) sites according to gene body location. Gray lines indicate numbers found in a gene-body location-matched background set of sites of equivalent number; (5-6) distribution of the relevant motif relative to m⁶A-Nanopore* (5) or ECT2–iCLIP (6) sites. Gray lines represent the distribution for the same gene-body location-matched set as derived in the panels above. * *Parker et al., 2020*; ** *Shen et al., 2016*; *** *Wei et al., 2018*.

*Figure 5 continued on next page*

*Figure 5 continued*

The online version of this article includes the following source data and figure supplement(s) for figure 5:

**Figure supplement 1.** Sources of motifs and generation of position weight matrices (PWMs).

**Figure supplement 2.** Motif logos generated from position weight matrices.

**Figure supplement 3.** Enrichment of RRACH variants around m6A and ECT2 sites.

**Figure supplement 4.** Uridines flanking DRACH result in additional motifs enriched at ECT2 iCLIP sites.

**Figure supplement 4—source data 1.** High quality image file.

GGAU as an alternative methylation site (*Anderson et al., 2018*). The possible roles of the U/Y-rich motifs in m6A deposition and ECT2 binding are analyzed in the following sections.

## Neighboring U/Ys result in enriched RRACH- and GGAU-derived motifs

We first noticed that several motifs retrieved around ECT2 crosslink sites by Homer constituted extended versions of **DRACH/GGAU** with *U*s upstream (e.g., *U*GAAC/*U*GGAU) or remnants of DR**ACH** with *U/Cs* (Ys) downstream (e.g., **ACU**CU). To test whether these motifs are indeed located adjacent to m6A, we examined their distribution and enrichment around ECT2 and m6A sites. The distributions showed a clear enrichment at m6A positions with a shift in the direction of the *U/Y*-extension (see *Figure 5D* for ACUCU and *Figure 5—figure supplement 4* for others). An enrichment over location-matched background sites close to ECT2-iCLIP sites was also apparent (see *Figure 5D* for ACUCU and *Figure 5—figure supplement 4* for others), further supporting that ECT2 preferentially crosslinks to uridines located in the immediate vicinity of DRACH (/GGAU). Thus, several enriched motifs around ECT2 crosslink sites are DRACH/GGAU-derived, and their detection in unbiased searches simply reflects a tendency of methylated DRACH/GGAU sites to be flanked by U/Ys.

## Nature of U/Y-rich motifs more distant from m6A sites

U/R-rich motifs without traces of adjacent DRACH (e.g., YUGUM, URUAY, URURU) showed a characteristic enrichment around, but depletion at, m6A sites. For some motifs, the enrichment was more pronounced 5′ than 3′ to m6A sites (see *Figure 5D* for URUAY and *Figure 5—figure supplement 4* for others). The distance between the site of maximal motif occurrence and the m6A site roughly coincided with the shift observed in ECT2 crosslink sites relative to m6A (*Figure 5A*, upper panel). Accordingly, these motifs were enriched exactly at ECT2 crosslink sites (see *Figure 5D* for URUAY and *Figure 5—figure supplement 4* for others), suggesting that they may constitute additional m6A-independent sites of interaction with ECT2. We also observed that the 3′ enrichment of YYYYY was asymmetric and closer to m6A than that of UUUUU/URURU/URUAY (*Figure 5—figure supplement 4*, second row from the top), indicating a preference for hetero-oligopyrimidine tracts immediately downstream the m6A site, as suggested by the 3′-enrichment of DRACUCU-type motifs as described above.

Taken together, these results suggest that *N6*-adenosine methylation preferentially occurs in DRACH/GGAU sequences surrounded by stretches of pyrimidines, with a preference for YYYYY (e.g., CUCU) immediately downstream, URURU (including URUAY) immediately upstream, and UUUUU/UNUNU slightly further away in both directions. The enrichment of ECT2 crosslink sites at these motifs, and the fact that the m6A-binding-deficient mutant of ECT2 (W464A) crosslinks preferentially to 3′-UTRs through its N-terminal IDR, indicates IDR-mediated binding to U/R- and Y-rich motifs around m6A.

## DRACH/GGAU motifs are determinants of m6A deposition at the site, while flanking U(/Y)-rich motifs are indicative of m6A presence and ECT2 binding

Since our analysis thus far uncovered several motifs of potential importance for m6A deposition and ECT2 binding, we employed machine learning to distinguish m6A and ECT2 iCLIP sites from random location-matched background sites using motif-based features. Importantly, the underlying classification model includes all motif features within the same model, allowing an evaluation of the importance of the motifs relative to each other. We used as features the number of matches to each

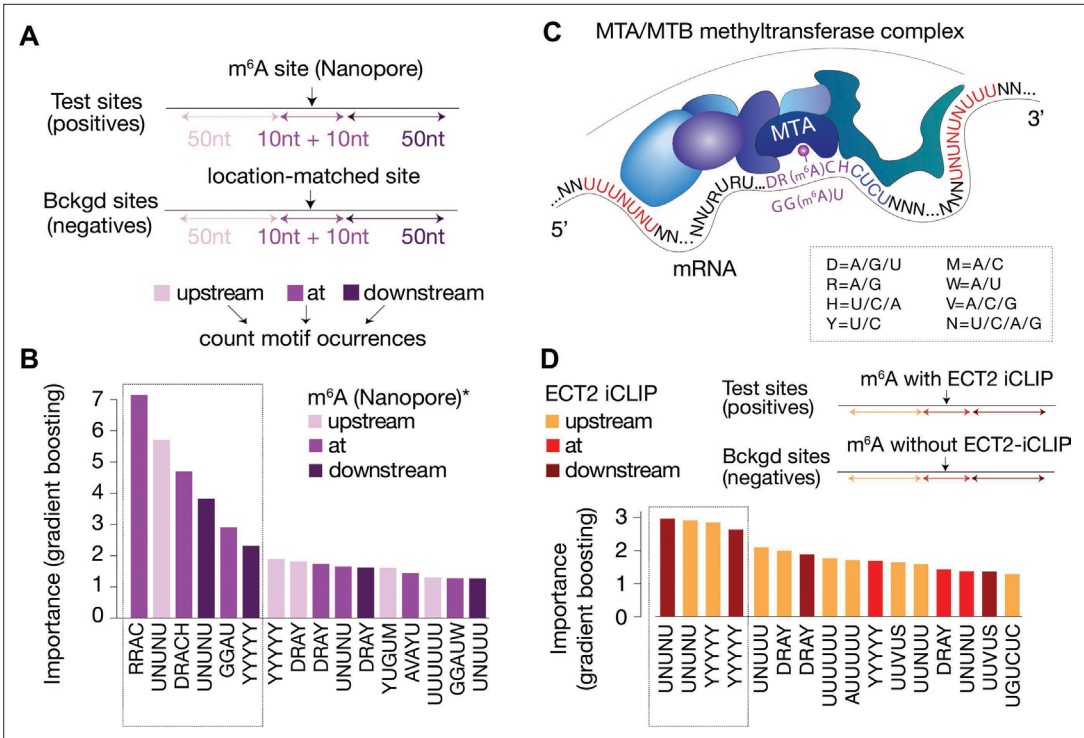

**Figure 6.** Distal U-rich motifs and at-the-site DRACH/GGAU are determinants for m⁶A deposition. (**A**) Diagram representing the strategy for machine learning model trained to distinguish m⁶A-Nanopore* sites from their respective gene-body location matched background sets. (**B**) Bar plots showing top 16 motif feature importance scores from the m⁶A model, ordered from left to right by importance. The dotted rectangle highlights motifs with outstanding importance compared to the rest. (**C**) Cartoon representing the most important motifs found at and around m⁶A sites. UPAC-IUB codes to define multiple nucleotide possibilites in one position are indicated. (**D**) Machine learning model trained to distinguish between m⁶A sites with and without ECT2 crosslink sites, and the resulting bar plot showing top 16 motif feature importance scores. Nucleotide distances for intervals, order and dotted box are as in **A**/**B**. * *Parker et al., 2020*.

The online version of this article includes the following figure supplement(s) for figure 6:

**Figure supplement 1.** Model performance receiver operating characteristic (ROC) curves for distinguishing sequence preferences of either m⁶A or ECT2-bound sites.

of the 48 motifs (*Figure 5—figure supplement 2*) in three distinct regions relative to the methylated site according to Nanopore sequencing (*Parker et al., 2020*), defined as position 0: 'at' [−10 nt; +10 nt], 'down' [−50 nt; −10 nt], or 'up' [+ 10 nt; +50 nt] (*Figure 6A*). The model involving all motifs could successfully distinguish the methylated sites from the background as indicated by an area under the receiver operating characteristic (ROC) curve (true positive rate versus false positive rate, area under the curve [AUC]) of 0.93, and even a reduced model incorporating only the top 10 features from the full model classified sites largely correctly (AUC = 0.86; *Figure 6—figure supplement 1*). The top 16 features ordered by importance from the full model confirmed that RRAC/DRACH or GGAU at the site was indicative of the presence of m⁶A (*Figure 6B*). Interestingly, U/Y-rich sequences (UNUNU and YYYYY in particular) flanking the site were also strongly indicative (*Figure 6B*). Some motifs showed a skew in their feature importance score, with UNUNU and YUGUM showing a preference to be upstream, and YYYYY downstream (*Figure 6B*), thus corroborating our previous observations (*Figure 6C*).

We used a similar modeling approach to identify non-m⁶A determinants of ECT2 binding, in this case comparing m⁶A sites within 10 nt distance of ECT2-iCLIP sites to m⁶A sites without ECT2-iCLIP sites nearby (AUC = 0.94, and AUC = 0.84 using only the top 10 features, *Figure 6—figure supplement 1*). In agreement with previous observations, this model showed flanking U/Y-rich sequences as the main determinants for ECT2 crosslinking (*Figure 6D*).

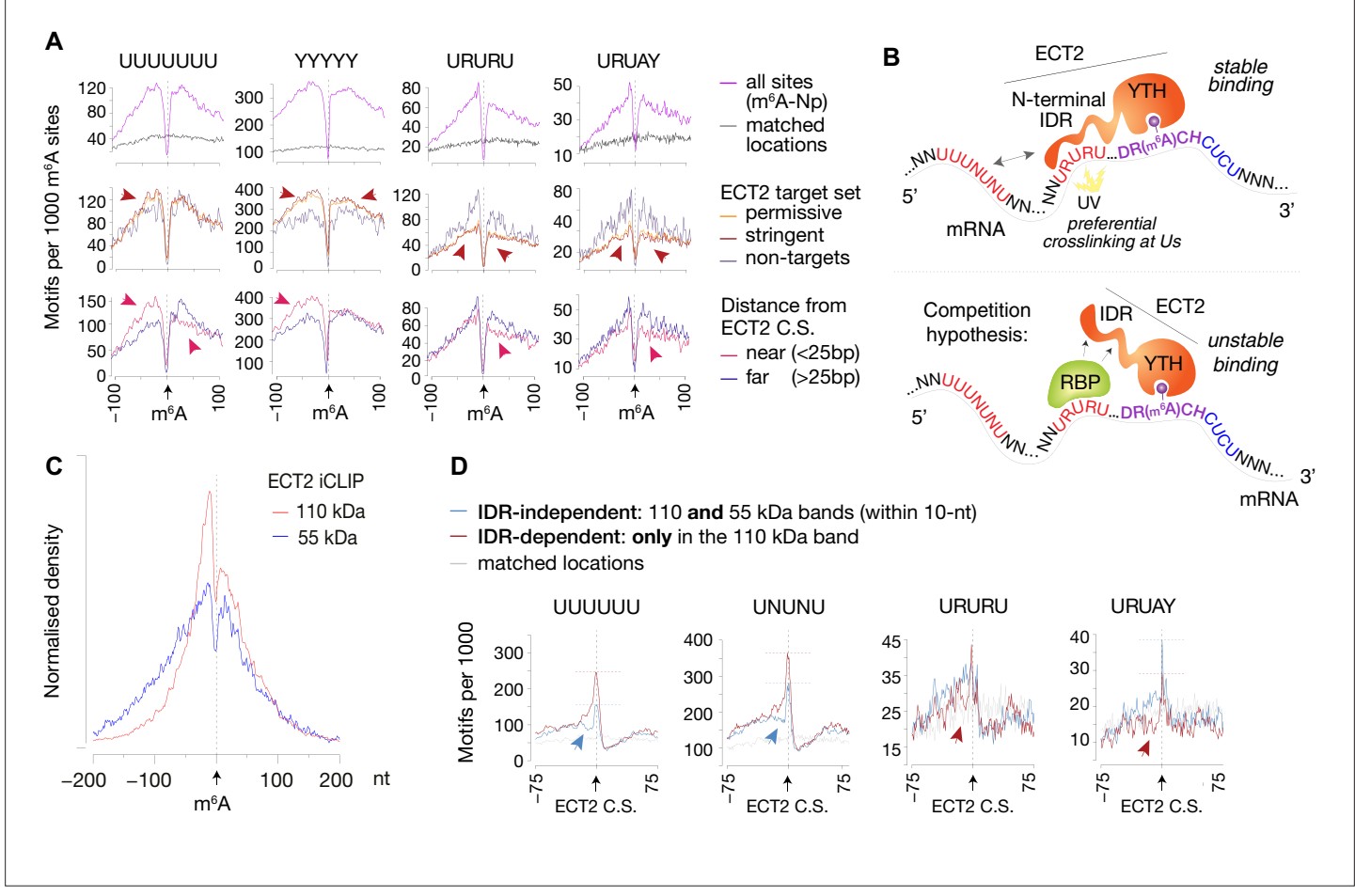

**Figure 7.** IDR-dependent binding of ECT2 to U-rich motifs 5' of m⁶A. (**A**) Top panels: Distance-based enrichment of motifs at and around m⁶A-Nanopore (Np, **Parker et al., 2020**) sites, plotted as motif counts per 1000 m⁶A sites (purple lines). Gray lines indicate the enrichment in a location-matched background set as in **Figure 5D**. Middle and bottom panels: sites are split according to whether they sit on ECT2 targets (middle), or to distance from the nearest ECT2 crosslink site (for ECT2-iCLIP targets only) (bottom). Additional motifs are shown in the **Figure 7—figure supplement 1**. (**B**) Cartoon illustrating the ECT2 IDR RNA-binding and competition hypotheses. (**C**) Normalized density of ECT2 iCLIP crosslink sites identified in the libraries corresponding to the 110- and 55-kDa bands (**Figure 3B**) at and up to +/-200 nt of m⁶A-Nanopore sites. (**D**) Motifs per 1000 ECT2-iCLIP crosslink sites (CS) split according to whether they are found in libraries from both 110-kDa and 55-kDa bands (IDR-independent'), or exclusively (distance > 10 nt) in the 110-kDa band ('IDR-dependent'). Gray lines indicate the enrichment in a location-matched background set as in **Figure 5D**. Additional motifs are shown in **Figure 7—figure supplement 2** and **Supplementary file 3**.

The online version of this article includes the following source data and figure supplement(s) for figure 7:

**Figure supplement 1.** Motif preferences around m⁶A sites according to ECT2 binding.

**Figure supplement 1—source data 1.** High quality image file.

**Figure supplement 2.** Dependency of the ECT2 intrinsically disordered region (IDR) for motif enrichment.

**Figure supplement 2—source data 1.** High quality image file.

## The U(-R) paradox: URURU-like sequences around m⁶A sites repel ECT2 binding, while U-rich sequences upstream enhance its crosslinking

To investigate the idea of URURU-like motifs as additional sites of ECT2 binding upstream of the m⁶A-YTH interaction site, we split Nanopore-m⁶A sites according to two criteria: (1) whether they occur in ECT2-target transcripts (both permissive and stringent sets analyzed separately), and (2) for ECT2 targets, whether there is an ECT2 crosslink site within 25 nt of the m⁶A site ('near') or not ('far'). Although there was no obvious differences between these categories for most of the motifs (**Supplementary file 3**, page 2), some U-rich sequences displayed distinctive features (**Figure 7A**, **Figure 7—figure supplement 1**) that can be summarized as follows. If a transcript has m⁶A *and* ECT2

sites in close proximity, it is (1) more likely to have UNUNU/UUUUU/YYYYY sequences upstream of the m⁶A site than targets with distantly located ECT2-binding sites or than non-ECT2 targets; (2) less likely to have UUUUU/URURU sequences downstream of the m⁶A site, possibly because ECT2 prefers CUCU-like sequences downstream; and (3) less likely to have URURU/URUAY-like motifs upstream of the m⁶A site. The latter observation is striking because for the specific subset of ECT2-bound m⁶A sites with URURU/URUAY upstream of m⁶A, these sequences tend to crosslink to ECT2, as seen by the enrichment spike at ECT2 crosslink sites (*Figure 5D*, *Figure 7—figure supplement 1*, bottom panels). Although these two results seem contradictory at first glance, they may be reconciled by a model in which a URURU/URUAY-binding protein would compete with ECT2 for binding adjacent to m⁶A. If that protein is absent, ECT2 may bind to the site, potentially via its IDR, to stabilize the low-affinity YTH-m⁶A interaction and crosslink efficiently due to the U-content. Conversely, if occupied by the alternative interacting protein, the site might repel ECT2 (see Discussion and *Figure 7B*).

## The N-terminal IDR of ECT2 is involved in preferential crosslinking at U-rich sequences and in URURU-repulsion immediately upstream m⁶A sites

We reasoned that insights into contacts between ECT2 and mRNA may be gained by analysis of the iCLIP libraries prepared with the 'YTH-mCherry' truncation devoid of the N-terminal IDR ('55-kDa band') compared to the full-length ECT2-mCherry ('110-kDa band') (*Figure 3*, *Figure 3—figure supplements 2–4*). Initial inspection of the distribution of ECT2 peaks relative to Nanopore-m⁶A sites showed that the 5′–3′ asymmetry observed with full-length ECT2 was largely reduced with the truncated protein (*Figure 7C*), as was the bias towards uridines (*Figure 7—figure supplement 2A*). These observations suggest that the IDR indeed is implicated in binding to U-rich regions upstream of m⁶A. We next split the full-length ECT2 iCLIP peaks according to whether they are present in libraries from both full-length and truncated forms ('IDR-independent') or exclusively in the full-length ('IDR-dependent') (distance >10 nt) and plotted the enrichment of the studied motifs relative to the crosslink site (*Figure 7D*, *Figure 7—figure supplement 2B*; *Supplementary file 3*, page 2). UUUUUU/UNUNU-like motifs were more enriched at and immediately upstream of IDR-dependent crosslink sites relative to the IDR-independent ones, supporting preferential crosslinking of the IDR to Us in this region. Remarkably, the exact opposite was true for URURU/URUAY motifs that showed modest depletion 5′ to IDR-dependent crosslink sites relative to their IDR-independent counterparts (*Figure 7D*). These observations are consistent with a model of an RNA-binding protein competing with the ECT2 IDR for interaction with upstream URURU/URUAY motifs (*Figure 7B*).

## Discussion
### Methodology for mapping protein-RNA interactions in plants

Our work establishes experimental and computational approaches to implement HyperTRIBE for unbiased and sensitive mapping of direct targets of RNA-binding proteins in plants. Two points are particularly relevant in this regard. First, the examples studied here show that stable transgenic expression of *Dm*ADARcd does not lead to detrimental phenotypes, perhaps because of the generally low editing proportions obtained in vivo. Second, the rigorous statistical approach developed to call editing sites makes HyperTRIBE powerful, despite the low editing proportions observed. We also note that ECT2 is well suited to verify that HyperTRIBE mostly recovers directly bound target RNAs because of the possibility to cross-reference the data with independently obtained m⁶A maps (*Parker et al., 2020*). The combination of iCLIP and HyperTRIBE for unbiased mapping of targets proved particularly attractive for at least two reasons. First, the convergence on overlapping target sets by orthogonal methods strengthens the confidence that the identified targets are biologically meaningful. Second, HyperTRIBE, especially with the novel computational approach for calling of editing sites (*Rennie et al., 2021*), offers higher sensitivity than iCLIP, while iCLIP is unmatched in providing information on binding sites within target RNAs. It is possible that better positional information on binding sites may be obtained from HyperTRIBE data using maximal editing proportions rather than statistical significance as the parameter to call editing sites. Indeed, recent work on the use of HyperTRIBE to identify targets of the RNA-binding protein MUSASHI-2 (MSI-2) in leukemic stem cells recovered the known MSI-2-binding site as enriched around editing sites in targets (*Nguyen et al., 2020*).

Nonetheless, our data shows that highly edited sites match the ADAR substrate consensus site better than lowly edited sites, suggesting that site proximity to ADAR is not the only determinant of editing proportions. Finally, our work also clearly indicates that FA-CLIP, now used in at least two studies involving YTH domain proteins (*Wei et al., 2018*; *Song et al., 2021*), is not a recommendable technique as it recovers many false positives and fails to include many genuine targets. Thus, with the possible exception of cases in which evidence for indirect association is specifically in demand, such as the recent study in human cells of mixed tailing of viral RNA by the cellular terminal nucleotidyl transferase TENT4 (*Kim et al., 2020*), FA-CLIP should not be used for identification of RNAs associating with a particular RNA-binding protein of interest.

## Core elements in m⁶A writing: DRACH, GGAU, and U/Y-rich motifs

Our analyses of motif enrichments around m⁶A and ECT2 crosslink sites clarify roles of previously reported motifs and uncover new motifs of importance in m⁶A writing and ECT2 binding. Since m⁶A is a prerequisite for ECT2 binding, any analysis of determinants of ECT2 binding must consider determinants of *N6*-adenosine methylation separately. Three conclusions stand out from our analysis in this regard. First, the major *N6*-adenosine methylation site is DRACH, consistent with conclusions from multiple other studies. Second, GGAU is a minor *N6*-adenosine methylation site, as seen by its enrichment directly at m⁶A sites. Third, m⁶A occurs in DRACH/GGAU islands embedded in U-rich regions. Such U-rich regions around m⁶A sites emerged from sorting of methylated from non-methylated transcripts by machine learning as being of similar importance for recognition of m⁶A-containing transcripts from sequence features as DRACH and GGAU at m⁶A sites, suggesting their implication in MTA/MTB-catalyzed adenosine methylation (*Figure 6C*). This, in turn, may also explain the pronounced 3'-UTR bias of m⁶A occurrence as extensive poly-U and poly-pyrimidine tracts are rare in coding regions (*Figure 5D*, second row on the right-most column; *Supplementary file 3*, page 1). As a special case in this context, our analyses suggest a simple explanation for the tendency of m⁶A to occur at stop codons. UAA and UGA correspond to DRA, increasing the frequency of occurrence of DRACH directly at stop codons (*Figure 5D*, second row on the left-most column), many of which have adjacent U-rich elements in the 3'-UTRs. We note that the observed pattern is in agreement with a role of the poly(U)-interacting proteins RBM15A/B associated with the mammalian methyltransferase complex in guiding methylation (*Patil et al., 2016*). Whether a similar mechanism operates in plants, potentially via the distant RBM15A/B homologue FPA (*Arribas-Hernández and Brodersen, 2020*), remains to be investigated.

## Reading of DR(m⁶A)CH in 3'-UTRs of target mRNAs by ECT2

It is a major conclusion of the present work that ECT2 binds to m⁶A predominantly in the DR(m⁶A)CH sequence context in vivo, consistent with reading of m⁶A written by the conserved nuclear MTA/MTB methyltransferase. This key conclusion refutes the claim by *Wei et al., 2018* that ECT2 binds to the supposedly plant-specific m⁶A-containing sequence motif URU(m⁶A)Y, and it thereby reconciles knowledge on m⁶A-YTHDF axes in plants specifically and in eukaryotes more broadly. The phenotypic similarity of plants defective in MTA/MTB writer and ECT2/ECT3/ECT4 reader function is now coherent with the locations of MTA/MTB-written m⁶A and ECT2-binding sites transcriptome-wide, and it is now clear that plants do not constitute an exception to the general biochemical framework for eukaryotic m⁶A-YTHDF function in which YTHDF proteins read the m⁶A signal written by the MTA/MTB methyltransferase.

## The role of U-rich motifs 5' to m⁶A sites in ECT2 binding: direct interaction of the IDR of ECT2 with mRNA

The pronounced protease sensitivity of IDRs, leading to limited proteolysis of ECT2 upon cell lysis after in vivo crosslinking, allowed us to extract information on the mode of ECT2-RNA binding from different observations, all converging on the conclusion that the IDR of ECT2 participates in RNA binding. First, RNA complexes with YTH-mCherry were 5'-labeled by polynucleotide kinase much more efficiently than RNA complexes with full-length ECT2-mCherry, indicating that the IDR limits accessibility to the 5' of bound mRNAs. Second, in contrast to the m⁶A-binding-deficient YTH^W464A^-mCherry truncation, the full-length ECT2^W464A^-mCherry mutant retained an enrichment of crosslink sites in 3'-UTRs. Third, crosslinks specific to the IDR (i.e., observed only with full-length ECT2-mCherry-RNA

complexes, but not with YTH-Cherry-RNA complexes), could be assigned and have two notable properties. They are mainly 5′ to m⁶A sites, and thereby cause a conspicuously asymmetric distribution of ECT2 crosslink sites around m⁶A sites, not seen with crosslinks to the YTH-mCherry fragment. In addition, the IDR-specific crosslinks are specifically enriched in U-rich elements of the type UUUUUU and UNUNU immediately upstream. Taken together, these observations suggest that the IDR of ECT2 participates in locating ECT2 to 3′-UTRs by association with U-rich elements. Thus, ECT2, and perhaps YTHDF proteins more generally given their highly similar YTH domains, appears to bind RNA through multivalent interactions among which the YTH domain is responsible for m⁶A binding, and the IDR is responsible for interaction with adjacent elements. We note that the notion of RNA interaction by IDRs has precedent (*Corley et al., 2020*), is consistent with the modest affinity of isolated YTHDF domains for m⁶A-containing oligonucleotides (*Patil et al., 2018*), and is reminiscent of the recent demonstration that transcription factors use their globular DNA-binding domains to recognize core sequence elements of promoters, and their IDRs to provide additional DNA contacts, contributing to specificity (*Brodsky et al., 2020*). Similarly, it is possible that diverging IDRs among YTHDF paralogs could confer target specificity via binding to distinct motifs in the vicinity of m⁶A sites, such that specific YTHDF-target mRNA repertoires could exist even for YTHDF proteins coexpressed in the same cells. Finally, we stress that although our data point to an important role of the IDR in RNA binding, it does not in any way suggest that this is the only function of the IDR, and protein-protein interactions involving the IDR are likely to be key to understanding YTHDF function molecularly.

## URUAY as sites of competitive interaction between ECT2 and other RNA-binding proteins

Despite the conclusions that URUAY does not contain m⁶A in *Arabidopsis*, and that ECT2 binds to DR(m⁶A)CH, our detailed analysis of sequence motifs enriched around m⁶A and ECT2 iCLIP crosslink sites shows that additional motifs, including URUAY, are likely to be implicated in m⁶A reading by ECT2, even if not directly. In contrast to other m⁶A-proximal, pyrimidine-rich sequences (e.g., UNUNU, YYYYY) that may be of importance for both m⁶A writing and ECT2 binding, URUAY appears to have ties more specifically to ECT2 binding thanks to three properties. (1) When present 5′ to m⁶A sites, it crosslinks to ECT2, suggesting that some part of the protein can be in contact with URUAY. (2) URUAY is more enriched close to m⁶A sites for which there is no evidence of ECT2 binding, suggesting that it weakens ECT2 binding. This latter point is also consistent with the distinction of ECT2-bound from non-ECT2-bound m⁶A sites by machine learning that did not find URUAY to be of importance for ECT2-bound sites. (3) The URUAY enrichment 5′ to ECT2 crosslink sites is observed only when crosslinks to both full-length protein and the YTH-mCherry fragment are considered (IDR-independent), but disappears when crosslinks specific to the full-length protein (IDR-dependent) are analyzed. Although these observations may be explained by multiple scenarios, we find a simple, yet at present speculative, model attractive: URUAY may be a site of competition between the IDR of ECT2 and another, as yet unknown, RNA-binding protein. Such a competing factor could in theory be another YTHDF protein using higher-affinity IDR-URUAY contacts than ECT2 to achieve competitive binding. Many other possibilities exist, however. For example, it is intriguing that URUAY resembles part of a Pumilio-binding site (*Hafner et al., 2010*; *Huh et al., 2013*) as it raises the tantalizing possibility of functional interaction between YTHDF and Pumilio proteins. In any event, the functional dissection of the URUAY element in m⁶A reading now constitutes a subject of major importance, emphasized by the broad conservation of its enrichment around m⁶A sites across multiple plant species, including rice (*Li et al., 2014a*; *Zhang et al., 2019*), maize (*Luo et al., 2020*; *Miao et al., 2020*), tomato (*Zhou et al., 2019*), and *Arabidopsis* (*Miao et al., 2020*).

## Materials and methods

**Key resources table**

| Reagent type (species) or resource | Designation | Source or reference | Identifiers | Additional information |
|---|---|---|---|---|
| Gene (*Arabidopsis thaliana*) | ECT2 | TAIR10 | AT3G13460 | *EVOLUTIONARILY CONSERVED C-TERMINAL REGION 2* |

| Reagent type (species) or resource | Designation | Source or reference | Identifiers | Additional information |
|---|---|---|---|---|
| Gene (*Arabidopsis thaliana*) | *ECT3* | TAIR10 | AT5G61020 | *EVOLUTIONARILY CONSERVED C-TERMINAL REGION 3* |
| Gene (*Arabidopsis thaliana*) | *ECT4* | TAIR10 | AT1G55500 | *EVOLUTIONARILY CONSERVED C-TERMINAL REGION 4* |
| Gene (*Drosophila melanogaster*) | ADAR Isoform N | Genebank, FlyBase, NCBI | CG12598 NM_001297862 | Adenosine deaminase acting on RNA |
| Strain (*Escherichia coli*) | DH5α | NEB | Cat. # 18258012 | MAX Efficiency DH5α Competent Cells |
| Strain (*Agrobacterium tumefaciens*) | GV3101 | **Koncz and Schell, 1986** | | |
| Genetic reagent (*A. thaliana*) | SALK_002225 C (*ect2-1*) | NASC | N657472 N2110120 | |
| Genetic reagent (*A. thaliana*) | *te234 (ect2-1/ect3-1/ect4-2)* | **Arribas-Hernández et al., 2018** | N2110132 | Donated to NASC and ABRC |
| Genetic reagent (*A. thaliana*) | *ECT2pro:FLAG-DmADAR$^{E488Q}_{cd}$-ECT2ter* | This paper (see Methods) | | Seed requests to pbrodersen@bio.ku.dk |
| Genetic reagent (*A. thaliana*) | *ect2-1/ECT2pro:ECT2-FLAG-DmADAR$^{E488Q}_{cd}$-ECT2ter* | This paper (see Methods) | | Seed requests to pbrodersen@bio.ku.dk |
| Genetic reagent (*A. thaliana*) | *te234/ECT2pro:ECT2-FLAG-DmADAR$^{E488Q}_{cd}$-ECT2ter* | This paper (see Methods) | | Seed requests to pbrodersen@bio.ku.dk |
| Genetic reagent (*A. thaliana*) | *ect2-1/ECT2pro: ECT2-mCherry-ECT2ter* | **Arribas-Hernández et al., 2018**; **Arribas-Hernández et al., 2020** | N2110839 N2110840 | Donated to NASC and ABRC |
| Genetic reagent (*A. thaliana*) | *ect2-1/ECT2pro: ECT2$^{W464A}$-mCherry-ECT2ter* | **Arribas-Hernández et al., 2018**; **Arribas-Hernández et al., 2020** | N2110841 N2110842 | Donated to NASC and ABRC |
| Genetic reagent (*A. thaliana*) | *ect2−1/ ECT2pro:3xHA-ECT2-ECT2ter* | This paper (see Methods) | | Seed requests to pbrodersen@bio.ku.dk |
| Genetic reagent (*A. thaliana*) | *ect2−1/ ECT2pro:3xHA-ECT2$^{W464A}$-ECT2ter* | This paper (see Methods) | | Seed requests to pbrodersen@bio.ku.dk |
| Genetic reagent (*D. melanogaster*) | Canton-S | Bloomington *Drosophila* Stock Center | BDSC:64,349 | Used to extract RNA and produce cDNA for cloning |
| Antibody | anti-FLAG (mouse monoclonal) | Sigma-Aldrich | A8592 | Used for WB (1:1000) |
| Antibody | anti-mCherry (rabbit polyclonal) | Abcam | ab183628 | Used for WB (1:1000) |
| Antibody | anti-HA (mouse monoclonal) | Abnova | 12CA5 | Used for WB (1:2000) |
| Antibody | RFP-Trap RFP Nanobody/V$_H$H coupled to agarose (recombinant, monoclonal) | ChromoTek | Cat. # rta-20 | Used for IP (20 µL of beads for 4 g of tissue in 6 mL of buffer) |
| Antibody | Anti-HA Affinity Matrix from IgG1 3 F10 (rat, monoclonal) | Roche | Cat. # 11815016001 | Used for IP (10 µL of beads for 500 mg of tissue in 750 µL of buffer) |
| Recombinant DNA reagent | pCAMBIA3300U (plasmid) | **Nour-Eldin et al., 2006** | | Used for cloning |
| Commercial assay or kit | pGEM -T Easy (plasmid and cloning kit) | Promega | Cat. # A1360 | Used for cloning |
| Commercial assay or kit | KAPAHiFi HotStart Uracil + Kit | Roche | Cat. # 7959079001 | Used for cloning |
| Commercial assay or kit | AccuPrime Supermix I | Invitrogen | Cat. # 12342–010 | Used for iCLIP library preparation |
| Peptide, recombinant protein | Uracil-DNA Glycosylase (USER enzyme) | NEB | Cat. # M5505L | Used for cloning |

| Reagent type (species) or resource | Designation | Source or reference | Identifiers | Additional information |
|---|---|---|---|---|
| Peptide, recombinant protein | Turbo DNase | Ambion | Cat. # AM2238 | Used for CLIP |
| Peptide, recombinant protein | RNase I | Ambion | Cat. # AM2294 | Used for CLIP |
| Peptide, recombinant protein | T4 Polynucleotide Kinase (PNK) | ThermoFisher Scientific | Cat. # EK0031 | Used for iCLIP library preparation |
| Peptide, recombinant protein | T4 RNA Ligase I, High Concentration | NEB | Cat. # M0437M | Used for iCLIP library preparation |
| Peptide, recombinant protein | Proteinase K | Roche | Cat. # 3115887001 | Used for iCLIP library preparation |
| Peptide, recombinant protein | Superscript III Reverse Transcriptase | Invitrogen | Cat. # 18080–093 | Used for iCLIP library preparation |
| Peptide, recombinant protein | CircLigase II ssDNA Ligase | Epicentre | Lucigen Cat. # CL9021K | Used for iCLIP library preparation |
| Peptide, recombinant protein | BamHI (Fast Digest) | ThermoFisher Scientific | Cat. # FD0054 | Used for iCLIP library preparation |
| Chemical compound, drug | cOmplete protease inhibitor cocktail | Roche | Cat. # 11697498001 | Used for CLIP |
| Chemical compound, drug | Protease inhibitor cocktail for plant cell extracts | Sigma | Cat. # P9599 | Used for CLIP |
| Chemical compound, drug | Glufosinate-ammonium (PESTANAL) | Sigma | Cat. # 45520 77182-82-2 | Used for selection of transgenic lines |
| Sequence-based reagent | Pre-adenylated adapter for iCLIP (3'-RNA linker) | *Huppertz et al., 2014* | L3-App | rAppAGATCGGAAGAGCGGTTCAG/ddC/ |
| Sequence-based reagent | iCLIP RT-primers (Two-part cleavable DNA adapters complementary to the 3' RNA linker) | *Huppertz et al., 2014* | Rt1clip-Rt12clip | Used for iCLIP library preparation (seq: ) |
| Sequence-based reagent | USER and site-directed mutagenesis primers | This paper (Appendix) | | Used for cloning. Sequences are in the Appendix |
| Sequence-based reagent | Primers for detection of point mutations | This paper (Appendix) | | Used for cloning. Sequences are in the Appendix |
| Software, algorithm | R | https://www.R-project.org/ | | Used for data analyses |
| Software, algorithm | hyperTRIBE**R** | *Rennie et al., 2021*; https://github.com/sarah-ku/hyperTRIBER; https://github.com/sarah-ku/targets_arabidopsis | | Used for calling significant ADAR-edited sites. Contact: sarah@binf.ku.dk |
| Software, algorithm | *trimmomatic* | *Bolger et al., 2014* | | Used for trimming RNAseq-reads |
| Software, algorithm | STAR | *Dobin et al., 2013* | | Used for mapping RNAseq-reads |
| Software, algorithm | Salmon | *Patro et al., 2017* | | Used for transcript quantification |
| Software, algorithm | SAMtools *mpileup* | *Li et al., 2009* | | Used to count nt-mismatches |
| Software, algorithm | *rtracklayer* | *Lawrence et al., 2009* | | Used to retrieve sequences |
| Software, algorithm | *ggseqlogo* | *Wagih, 2017* | | Used to generate motif logos |
| Software, algorithm | *Hmisc* | https://github.com/harrelfe/Hmisc/ | | Used for expression-based binning |
| Software, algorithm | *fastqc* | https://www.bioinformatics.babraham.ac.uk/projects/fastqc/ | | Used for quality control |
| Software, algorithm | *cutadapt* | *Martin, 2011* | | Used for trimming of iCLIP reads |
| Software, algorithm | *flexbar* | *Roehr et al., 2017* | | Used for demultiplexing iCLIP reads |

| Reagent type (species) or resource | Designation | Source or reference | Identifiers | Additional information |
|---|---|---|---|---|
| Software, algorithm | PureCLIP | *Krakau et al., 2017* | | Used for calling iCLIP peaks |
| Software, algorithm | *GenomicRanges* | *Lawrence et al., 2013* | | Used to retrieve short sequences |
| Software, algorithm | 'Distributions of motifs per 1,000 sites over distance' | This paper https://github.com/ sarah-ku/targets_ arabidopsis | | Used to calculate motif distributions around m⁶A/iCLIP. Contact: sarah@ binf.ku.dk |
| Software, algorithm | *ggplot2* | https://ggplot2. tidyverse.org | | Used to generate plots |
| Software, algorithm | *bedtools* | *Dale et al., 2011*; *Quinlan and Hall, 2010* | | Used to filter and clean iCLIP data |
| Software, algorithm | Homer | *Heinz et al., 2010* | | Used for de novo motif discovery |
| Software, algorithm | FIMO | *Grant et al., 2011* | | Used to detect motif occurrences |
| Software, algorithm | *gbm* | https://github.com/gbm-developers/ gbm | | Used for random forest analysis |
| Software, algorithm | *pROC* | *Robin et al., 2011* | | Used to estimate predictive score of RF |
| Software, algorithm | IGV (Integrative Genomics Viewer) | *Robinson et al., 2011* | | Used to show genomic data |

All data analyses were carried out using TAIR 10 as the reference genome and Araport11 as the reference transcriptome. Unless otherwise stated, data analyses were performed in R (https://www.R-project.org/) and plots generated using either base R, IGV (for genomic data) (*Robinson et al., 2011*), or *ggplot2* (https://ggplot2.tidyverse.org).

## Definitions of experiment, biological replicates, and technical replicates

We use the term 'biological replicate' in the following way: plants were grown at the same time, under the same conditions, but in separate plates. Each sample replicate contains pools of seedlings prepared in such a way that no two replicates contain seedlings grown on the same plates. This sampling ensures that plate-to-plate variation in growth conditions, if any, will have an effect on measurements of gene expression within a single genotype, and hence minimize the risk that any differences due to such variation are called as significant in comparisons between genotypes. 'Technical replicates' are understood to be independently conducted measurements using the same technique on the same biological material (e.g., on one biological replicate as defined above). Technical replicates were not carried out in this study, and the term 'replicate' refers to biological replicate as defined above. In our definition, an 'experiment' results in generation and comparison of measurements arising from multiple biological replicates of different biological entities, in the present case often *Arabidopsis* seedlings differing in genotype with respect to the genes *ECT2*, *ECT3,* and *ECT4.* Thus, repetition of an experiment in our definition entails generation and analysis of the required biological replicates at different points in time.

## Plant material

All lines used in this study are in the *Arabidopsis thaliana* Col-0 ecotype. The mutant alleles or their combinations – *ect2-1* (SALK_002225) (*Arribas-Hernández et al., 2018*; *Scutenaire et al., 2018*; *Wei et al., 2018*), *ect3-1* (SALKseq_63401), *ect4-2* (GK_241H02), and *ect2-1/ect3-1/ect4-2* (*te234*) (*Arribas-Hernández et al., 2018*) – have been previously described. The transgenic lines expressing *ECT2pro:ECT2-mCherry-ECT2ter*, *ECT2pro:ECT2^{W464A}-mCherry-ECT2ter*, *ECT2pro:3xHA-ECT2-ECT2ter*, or *ECT2pro:3xHA-ECT2^{W464A}-ECT2ter* in the *ect2-1* background have also been described or generated by floral dip in additional mutant backgrounds using the same plasmids and methodology (*Arribas-Hernández et al., 2018*; *Arribas-Hernández et al., 2020*).

## Growth conditions

Seeds were surface-sterilized by 2 min incubation in 70% EtOH plus 10 min in sterilizing solution (1.5% NaOCl, 0.05% Tween-20) and 2 $H_2O$ washes. After 2–5 days of stratification at 4 °C in darkness, seeds were germinated and grown on plates containing Murashige and Skoog (MS)-agar medium (4.4 g/L MS, 10 g/L sucrose, 10 g/L agar) pH 5.7 at 20 °C, receiving ~70 µmol m$^{-2}$ s$^{-1}$ of light in a 16 hr light/8 hr dark cycle as default. For HyperTRIBE and iCLIP experiments, the plates were placed vertically to facilitate root harvesting. MS-agar media for HyperTRIBE T2 seedlings was supplemented with 7.5 mg/L of glufosinate ammonium (Sigma) to select plants expressing the ADAR-containing transgenes. To assess phenotypes of adult plants, ~8-day-old seedlings were transferred from horizontal MS plates (4.4 g/L MS, 10 g/L sucrose, 8 g/L agar; pH 5.7) to soil and maintained in Percival incubators under 16 hr light/8 hr dark cycles, 21 °C day/18°C night temperature, and ~100 µmol m$^{-2}$ s$^{-1}$ light intensity. We used Philips fluorescent tubes TL-D 90 De Luxe 36 W as light source.

## Generation of transgenic lines for HyperTRIBE

We employed USER cloning (*Bitinaite and Nichols, 2009*) to generate *ECT2pro:ECT2-FLAG-DmADAR$^{E488Q}$cd-ECT2ter* and *ECT2pro:FLAG-DmADAR$^{E488Q}$cd-ECT2ter* constructs in pCAMBIA3300U (pCAMBIA3300 with a double PacI USER cassette inserted between the *Pst*I-*Xma*I sites at the multiple cloning site; *Nour-Eldin et al., 2006*). Fragments containing *ECT2* gDNA sequences were amplified by PCR (KAPA HiFi Hotstart Uracil + ReadyMix, Roche) from plasmids previously generated in our lab (*Arribas-Hernández et al., 2018*). The *FLAG-DmADAR$^{E488Q}$cd* fragment was produced in the same way using a pGEM-T Easy (Promega) plasmid containing FLAG-*DmADAR$^{E488Q}$cd* as template, previously subcloned to introduce the E488Q hyperactive mutation by site-directed mutagenesis (QuickChange, Agilent Technologies) with primers LA729-LA730 (Phusion HF DNA Polymerase, NEB). The E488Q mutation was detected by *Nla*III (ThermoFisher) digestion of the PCR reaction (DreamTaq, ThermoFisher) obtained with primers LA660-LA735. Of note, the *FLAG* and *DmADARcd* sequences had been previously glued together by USER cloning to produce *AGO1pro:FLAG-DmADARcd-AGO1ter* in pCAMBIA3300U for unrelated purposes (unpublished work), and subsequently amplified by PCR with primers LA696-615 for introduction into pGEM-T Easy. To build *AGO1pro:FLAG-DmADARcd-AGO1ter* in the first place, the catalytic domain of the ADAR deaminase isoform N (Y268-E669) was amplified from cDNA of *D. melanogaster* Canton-S wild-type flies and larvae with USER primers MVUSER12-22. The rest of the fragments were amplified from *pCAMBIA3300U AGO1pro:FLAG-AGO1-AGO1ter* (*Arribas-Hernández et al., 2016*) with primers MVUSER1-11 and MVUSER23-6.

USER primers to amplify all fragments were designed to create overhangs compatible with either the *Pac*I USER cassette present in the pCAMBIA3300U plasmid or the flanking sequences of the neighboring fragments. All primer sequences, their combinations to produce PCR fragments, and the arrangement of the fragments for USER cloning can be found in Appendix 1.

Kanamycin-resistant colonies of *Escherichia coli* DH5α (NEB) transformed with the constructs were analyzed by restriction digestion and sequencing prior introduction of the plasmids in *Agrobacterium tumefaciens* GV3101 (*Koncz and Schell, 1986*) for plant transformation.

*Arabidopsis* stable transgenic lines were generated by floral dip transformation (*Clough and Bent, 1998*) of Col-0 WT, *ect2-1*, or *te234*, and selection of primary transformants (T1) was done on MS-agar plates supplemented with glufosinate-ammonium (Sigma) (10 mg/L). We selected five independent lines of each type based on segregation studies (to isolate single T-DNA insertions), phenotypic complementation (in the *te234* background), and transgene expression levels assessed by FLAG western blot.

## Western blotting

Protein extraction from 10-day-old seedlings and western blotting with FLAG, HA, and mCherry antibodies were done as previously described (*Arribas-Hernández et al., 2018*). Loading was documented by amido black, Coomassie, or Ponceau staining of the total protein on the membrane.

## RNA extraction and library preparation for HyperTRIBE

We extracted total RNA from manually dissected root tips and apices (removing cotyledons) of five independent lines (10-day-old T2 seedlings) of each of the lines used for ECT2-HT to use as biological

replicates. The tissue was flash-frozen in liquid nitrogen and ground into a fine powder using liquid nitrogen-cooled adaptors in a tissue homogenizer. For RNA extraction, we added 1 mL of TRI Reagent (Sigma) to the frozen tissue (<100 mg), mixed quickly by vortexing, added 0.2 mL of chloroform, and separated the two resulting phases by vigorous shaking and 10 min centrifugation at 4 °C. The RNA was then precipitated from the aqueous phase for 30 min at room temperature with 1 volume of isopropanol. RNA pellets were solubilized in 300 µL of $H_2O$ to remove polysaccharides through a mild precipitation by addition of 1/10 vol. 99% EtOH and 1/30 vol. of 3 M NaOAc (pH 5.2) and incubation on ice for 30 min. After 15 min of full-speed centrifugation at 4 °C to pellet polysaccharides, we re-precipitated the RNA from the supernatant with 2,5× vol. 99% EtOH and 1/10 vol. of 3 M NaOAc (pH 5.2), washed the pellet two times with 70% EtOH, and resuspended in 20–40 µL of $H_2O$. This highly pure total RNA was then used to produce mRNA libraries through enrichment of mRNA with oligo(dT) beads (18-mers), random fragmentation, cDNA synthesis with random hexamers, custom second-strand synthesis (Illumina), terminal repair, A-ligation and sequencing adaptor ligation, size selection (250–300 bp insert), and PCR enrichment. The libraries were prepared and sequenced (Illumina PE150, Q30 ≥ 80%) as a service from Novogene.

The entire HyperTRIBE experiment was done once.

## HyperTRIBE data analysis

Significant differentially edited sites between *ECT2-FLAG-ADAR* (fusion) and *FLAG-ADAR* (control) samples for ECT2 HyperTRIBE (ECT2-HT) were called according to the hyperTRIBE**R** pipeline (*Rennie et al., 2021*). First, reads were trimmed using *trimmomatic* (*Bolger et al., 2014*) and mapped to the *Arabidopsis* genome (TAIR10) using STAR (*Dobin et al., 2013*), according to parameters suggested in a previous HyperTRIBE analysis (*Xu et al., 2018*). All HyperTRIBE samples were also quantified using Salmon (*Patro et al., 2017*), with appropriate settings for pair-end sequencing and non-stranded library setup and based on the transcriptome for Araport11 (*Cheng et al., 2017*) with manual addition of the *FLAG-ADAR* sequence. A custom Perl script based on SAMtools *mpileup* (*Li et al., 2009*) returned base counts for all positions where there is a mismatch from the reference in at least one sample. For running the hyperTRIBE**R** analysis pipeline, we specified that any tested position must have a putative edit in at least four of the five replicates in the *ECT2-FLAG-ADAR* samples (three of four in the case of roots since one of these samples, 'L3,' was deemed as low quality and subsequently removed from the significance calling pipeline). Significant hits (adjusted p-value<0.01 and $\log_2FC > 1$) were further filtered as follows: (1) hits that did not correspond to an A-to-G change (or a T-to-C change for the negative strand), (2) hits that were likely SNPs arising specifically in either the *ECT2-FLAG-ADAR* or *FLAG-ADAR* line manifesting in an editing proportion at or close to 1, and (3) hits where the coverage of tags at the edit base over the *ECT2-FLAG-ADAR* were fewer than 10 reads. Specific scripts for the analysis of ECT2-HT data can be found at https://github.com/sarah-ku/targets_arabidopsis.

Editing proportions were calculated as G/(A + G) (alternatively C/(U + C) for the negative strand) for all significant sites, averaged over all samples, separately for the *ECT2-FLAG-ADAR* and *FLAG-ADAR* samples. Significant sites were annotated to genes from Araport11, prioritizing the gene with the highest expression (annotated TPMs are based on Salmon quantifications of *FLAG-ADAR* control samples only) in the given tissue in the case of multiple possibilities. Possible transcripts were subsequently ordered by expression, along with gene body location along the transcript (5'-UTR, CDS, 3'-UTR).

Principal component analysis was carried out on the raw editing proportions per sample for all sites with significant evidence of editing.

For the comparison of sites between aerial tissues and roots, genes defined as commonly expressed in both types of tissues were considered in all gene-based comparisons. For significant editing site-based comparison, we directly compared sites that were common and significant to both.

To calculate correlations between editing proportions and *FLAG-ADAR* expression levels among lines, transcripts per million (TPM) mapping to *FLAG-ADAR* were extracted from quantifications from Salmon (*Patro et al., 2017*) and correlated with the raw editing proportions per sample, separately for the fusion and control samples. Background correlation estimates were calculated by first scrambling the order of the *FLAG-ADAR* TPM vector.

For motif identification at significant ECT2-HT sites, all sequences for bases at and -/+ 2 nt of the significant editing positions were derived from TAIR10 using the R package *rtracklayer* (*Lawrence et al., 2009*) in either aerial tissues or roots. A matrix of nt frequencies (A, C, G, or U) was generated, and the R package *ggseqlogo* (*Wagih, 2017*) was used to generate the final motif.

For the calculation of editing proportions as a function of the proportion of cells coexpressing ECT2, we first downloaded the expression matrix based on a total of 4727 individual cells from scRNA-seq in roots from *Denyer et al., 2019*. To estimate the relationship between coexpression of target genes with ECT2 and their average editing proportions, the expression matrix was used to calculate coexpression for each target gene as follows: (# cells expressing ECT2 AND target gene) / (# cells expressing target gene).

These proportions were then split into groups and plotted against the maximum editing proportions from HyperTRIBE in the containing genes.

## Comparative analysis of target sets and their expression bias

For expression binning, $\log_2$(TPM +1) values for all expressed genes in either aerial tissues, roots, or combined were split into nine bins of increasing expression, using the cut() function from the R package *Hmisc* version 4.5-0 (https://github.com/harrelfe/Hmisc/). For the proportion of target genes in every expression bin, we calculated the proportion of genes in each set (ECT2 HT/iCLIP-targets or nontargets, ECT2 FA-CLIP; *Wei et al., 2018*) or m6A sets (*Parker et al., 2020*) falling into each expression bin out of the total number of genes in that bin. To demonstrate expression biases in unsupported ECT2-HT target genes, the genes were further split according to whether or not they had support from m6A (Nanopore, miCLIP [*Parker et al., 2020*] and m6A-seq [*Shen et al., 2016*]).

## CLIP experiments and iCLIP library preparation

In vivo UV crosslinking of 12-day-old seedlings and construction of iCLIP libraries were optimized for ECT2-mCherry from the method previously employed for *Arabidopsis* GRP7-GFP (*Meyer et al., 2017*; *Köster and Staiger, 2020*) as follows. Crosslinked plant tissues (see details below) were finely ground in liquid nitrogen with mortar and pestle, homogenized in iCLIP buffer (50 mM Tris-HCl pH 7.5, 150 mM NaCl, 4 mM $MgCl_2$, 5 mM DTT, 1% SDS, 0.25% sodium deoxycholate, 0.25% Igepal) supplemented with protease inhibitors (4 mM PMSF, 1 tablet/10 mL of Complete Protease Inhibitor Cocktail [Roche], and 1/30 vol. of Protease Inhibitor Optimized for Plant Extracts [Sigma P9599]), and cleared by centrifugation and filtration (0.45 µm pore) of the supernatant. RNP-complexes were then immunopurified with beads coupled to anti-RFP nanobodies (ChromoTek RFP-Trap in our case) for 1 hr at 4 °C under constant rotation. In particular, we used 20 µL of beads for 4 g of tissue in 6 mL of iCLIP buffer for every replicate. After thorough washes with RIP-Wash Buffer (2 M urea, 50 mM Tris-HCl pH 7.5, 500 mM NaCl, 4 mM $MgCl_2$, 2 mM DTT, 1% SDS, 0.5% sodium deoxycholate, 0.5% Igepal), RNP-complexes attached to the beads were subjected to treatment with DNase (Turbo DNase [Ambion], 4 U/100 µL) and RNase I (Ambion, 1 U/mL) at 37 °C for 10 min, dephosphorylation of RNA 3′ ends (PNK [ThermoFisher] in pH 6.5 buffer), and 3′ RNA linker ligation (L3-App linker [*Huppertz et al., 2014*] and NEB HC RNA Ligase) at 16 °C overnight. RNA was radioactively labeled at the 5′ end by PNK-mediated phosphorylation using γ-$^{32}$P-ATP (20 min at 37 °C). The labeled RNP complexes were subjected to SDS-PAGE and blotting on a nitrocellulose membrane (Protran BA-85). Pieces of membrane containing a size range of RNA species bound to the protein (a smear above the expected molecular weight localized by autoradiography) were excised and subjected to proteolysis (200 µg of Proteinase K [Roche] in 200 µL of PK buffer [100 mM Tris-HCl pH 7.4, 50 mM NaCl, 10 mM EDTA] for 20 min at 37 °C) to release RNA bound to small peptides. The RNA was then purified with TRI-Reagent (Sigma) and used to prepare sequencing libraries through the following steps: reverse transcription (Superscript III, Invitrogen) using a two-part cleavable DNA adapter complementary to the 3′ RNA linker as primer, gel purification and size selection of cDNA (high, 120–200 nt; medium, 85–120 nt; low, 70–85 nt), circularization (CircLigase II Epicentre), relinearization (BamHI), and PCR amplification (AccuPrime Supermix I, Invitrogen). All steps were performed as described by *Huppertz et al., 2014*, and the amount of cycles in the final PCR was optimized to the amount of cDNA in each sample.

Notice that we introduced a few modifications in the original protocol (*Köster and Staiger, 2020*) to account for (1) low abundance of ECT2 compared to AtGRP7. To obtain enough RNA, we increased

the crosslinking energy and irradiated 12-day-old seedlings with 2000 mJ/cm$^2$ of 254 nm UV light, harvesting roots and shoots (4 g of tissue per replicate) to maximize the amount of purified ECT2-mCherry. (2) ECT2 sensitivity to proteolysis. We did not pre-clear the lysates to reduce the incubation time, and we used high amounts of protease inhibitors during immunoprecipitation. (3) High molecular weight of ECT2-mCherry. Due to the size of the protein, we required longer electrophoresis time and cooling (3 hr at 180 V with the tank on ice). (4) Different RNA-binding capacity of ECT2. Based on trials, we decided to adjust the RNase I treatment to 1 U/mL, incubating for 10 min at 37 °C (5 μL of RNase I [Ambion, 100 U/μL pre-diluted 1:5000] in 100 μL).

Of note, the conditions indicated here were specifically used for library preparation. Although we used the same conditions as default for CLIP experiments to assess ECT2 RNA-binding capacity, ECT2 sensitivity to proteolysis and ECT2-bound RNA sensitivity to RNase treatment, variations in buffer composition, incubation time, concentration of protease inhibitors, and/or RNase I are specified in the corresponding figure legends where necessary.

The entire iCLIP-seq experiment including three replicates of each group was done once.

## iCLIP data analysis and peak calling

Sequenced reads from all samples were investigated after each processing step with *fastqc* 0.11.5 (https://www.bioinformatics.babraham.ac.uk/projects/fastqc/). Adapters at the 3′ end were trimmed using *cutadapt* version 1.16 (*Martin, 2011*). The demultiplexing of the samples was performed using *flexbar* 3.4.0 with the -bk parameter to conserve the barcode information for further steps (*Roehr et al., 2017*). Reads with a length below 24 nucleotides were discarded. Barcodes were trimmed and saved to the *read_id* field. Processed reads were mapped to the TAIR10 genome with STAR version 2.6.0a  allowing a maximum of two mismatches and soft clipping only at 3′ end (*Dobin et al., 2013*). PCR duplicates were removed by grouping the reads by their mapping start position. Reads with the identical start position and random barcode were removed from the samples (Python3 and pybedtools). The peak calling of uniquely mapped reads was done using PureCLIP 1.0.4, choosing the second peak-shape option to allow more broader peaks to be called (*Krakau et al., 2017*).

For consistency with the ECT2-HT datasets, the ECT2-iCLIP datasets were annotated using the hyperTRIBE**R** annotation (*Rennie et al., 2021*), using quantifications based on the average of roots and aerial tissues from *FLAG-ADAR* samples in ECT2-HT (to reflect that the ECT2-iCLIP data is based on whole seedlings).

To calculate the proportion of sites falling at each nucleotide, nucleotide sequences from the reference genome were first obtained from site coordinates for ECT2 iCLIP/m$^6$A-Nanopore/ m$^6$A-miCLIP using the R packages *GenomicRanges* (*Lawrence et al., 2013*) and *rtracklayer* (*Lawrence et al., 2009*). Nucleotide proportions were plotted using ggplot2 (https://ggplot2.tidyverse.org).

## Analysis of publicly available data

Single-cell expression data and marker genes associated with 15 clusters annotated to cell types in roots were downloaded from *Denyer et al., 2019*. Single-nucleotide resolution locations of m$^6$A sites (defined according to Nanopore or miCLIP) were downloaded from *Parker et al., 2020*. Intervals defining m$^6$A locations based on m$^6$A-seq were downloaded from *Shen et al., 2016*, and intervals defining locations of ECT2-bound sites as determined by FA-CLIP were downloaded from *Wei et al., 2018*. For consistency with HyperTRIBE and ECT2-iCLIP, all sets of m$^6$A or ECT2-bound sites were gene annotated using the hyperTRIBE**R** pipeline, based on genes and transcripts from Araport11.

## Motif discovery

To remove redundancy after ECT2-iCLIP peak calling, directly adjacent peaks (crosslink sites) were grouped together and only the peak with the highest pureCLIP score (dominant) was kept. The called peak position (1 nt resolution) was extended by 4 nt up- and downstream to define a 'collapsed crosslink site' (CSS) with length 9 nt. The center position marks the dominant called peak. The extension of the peak positions was computed using *bedtools* version 2.27.1 (*Quinlan and Hall, 2010*; *Dale et al., 2011*). The collapsed ECT2-iCLIP crosslink sites and m$^6$A-Nanopore sites (*Parker et al., 2020*) were used to find motifs significantly enriched by Homer (*Heinz et al., 2010*) using a variety of window sizes, settings, and backgrounds. Motifs resulting from Homer searches were collated manually, and a range of variants of the consensus motif RRACH (e.g., RACH, DRAY, DRACH, URACH, DRACG) were

also added to the list, as well as various combinations of U-rich sequences (e.g., UUUUU, UNUNU, etc.), specific motifs found to be of interest in scientific literature (e.g., URUAY [*Wei et al., 2018*], GGAU [*Anderson et al., 2018*]), and extra motifs that appeared of potential interest from manually browsing with IGV (*Robinson et al., 2011*) the sequence in the vicinity of iCLIP peaks (e.g., YYYYY, DRACUCU). This resulted in a final list of 48 motifs for further analysis.

## Motif analysis

For each of the 48 motifs compiled from multiple sources, a custom PWM was generated based on local sequence frequencies around ECT2-iCLIP peaks and used as input to FIMO 5.1.1 (*Grant et al., 2011*) to detect genome-wide occurrences. To generate PWMs, we used the formula $PWM_{b,j} = \log_2 [p(b,j)/p(b)]$, where $p(b)$ is the background frequency of each nucleotide (see further down), and $p(b,i)$ is the frequency of the nucleotides in each position $j$. We also included an extra small frequency count in the calculation to account for potential uncertainty in redundancies. In order to account for location-specific sequence contexts (typically 3'-UTR), each site from iCLIP or m6A (*Parker et al., 2020*) sets was assigned a random 'matched background' site, in a non-target gene, at the same relative location along the annotated genomic feature of the site (5'-UTR, CDS or 3'-UTR), according to a resolution of 10 bins per feature. Logos for all motifs were generated using the R packages ggplot2 (https://ggplot2.tidyverse.org) and *ggseqlogo* (*Wagih, 2017*). To run the calculated PWMs through FIMO, we specified background letter frequencies (A: 0.273, C: 0.165, G: 0.173, U: 0.389), a threshold of 0.05, and scanning across the full TAIR10 *Arabidopsis* genome. Sites were further filtered downstream to have a score of at least 4 – in the vast majority of cases corresponding to an exact match the (short) motif.

Distributions of motifs per 1000 sites over distance, centering on ECT2-iCLIP or m6A sites and the respective matched backgrounds, were generated using a custom R-script (https://github.com/sarah-ku/targets_arabidopsis) based on overlaps using *GenomicRanges* (*Lawrence et al., 2013*). At any given shift from the peak set, the raw number of overlaps of the motif (at any point) was calculated and normalized to give a motif count per 1000 peaks. To adjust for the potential for downstream regions overshooting the end of the 3'-UTR, at each given distance only sites that continue to overlap an annotated gene (Araport11) are counted. For some analyses, peak sets were further split according to IDR-dependency or target status as indicated.

To calculate motif enrichment over the gene body, motifs were first annotated a value according to their relative position within the gene body regions: 5'-UTR, CDS or 3'-UTR. In order to account for over-representation of counts within the CDS, due to greater sequence coverage within transcript annotations, a random background set of 10 million positions were generated from the transcript annotation file and annotated in the same way as the motif locations to obtain an expected distribution of all positions over the gene body regions. This *E*xpected distribution was used to normalize the *O*bserved distribution of each motif, and O/E values were plotted as a metagene plot over the gene. An enrichment of 1 suggests that the motif is neither over- or under-represented at that location.

## Random forest analysis (machine learning)

Called positions from either Nanopore m6A data (*Parker et al., 2020*) or ECT2 iCLIP were first reduced to remove redundant regions of multiple peaks within the same window, then paired with matched background sets (described above). Windows representing 'at' (±10 nt) the motif together with adjacent upstream 'up' and downstream 'down' windows of length 50 nt (resulting in total window sizes of 120 nt) around each position were annotated according to the number of each of the motifs overlapping (truncated at 10), and the final data set normalized. To create a held-out set, 1/5th of the peaks were removed from the set, and the other 4/5th were used to build a random forest model using gradient boosting (R package *gbm* version 2.1.8; https://github.com/gbm-developers/gbm), with settings specifying a shrinkage of 0.05, an interaction.depth of 6, cv.folds = 5, and n.trees = 2000. For each model (m6A Nanopore-based or ECT2 iCLIP-based), importance scores were extracted from the model and the top features were selected. The held-out data was further used to estimate the predictive score of the model by calculating the AUC (R package *pROC*; *Robin et al., 2011*). Two further models were run – one involving the top 10 features from the full feature model, and (only for the m6A-Nanopore set-up) one involving features from

only DRACH and UNUNU (equating to six features in total), and AUC values were calculated and compared to that of the full feature model.

## Acknowledgements

We thank Lena Bjørn Johansson and Phillip Andersen for technical assistance in the construction of transgenic lines, and Theo Bølsterli, René Hvidberg Petersen, and their teams for plant care. Kim Rewitz is thanked for providing the *Drosophila* larvae and flies used for cDNA extraction to clone *Dm*ADARcd. We acknowledge Maria Louisa Vigh for cloning of FLAG-*Dm*ADARcd, Katja Meyer and Kristina Neudorf for support during iCLIP library construction in Bielefeld, and Simon Bressendorff and Mathias Tankmar for experimental support.

## Additional information

### Funding

| Funder | Grant reference number | Author |
| --- | --- | --- |
| H2020 European Research Council | ERC-2016-COG 726417 | Peter Brodersen |
| Independent Research Fund Denmark | 9040-00409B | Peter Brodersen |
| European Molecular Biology Organization | STF 7614 | Laura Arribas-Hernández |
| Deutsche Forschungsgemeinschaft | STA653/14-1 | Dorothee Staiger |
| H2020 European Research Council | 638173 | Robin Andersson |
| Independent Research Fund Denmark | 6108-00038B | Robin Andersson |

The funders had no role in study design, data collection and interpretation, or the decision to submit the work for publication.

### Author contributions

Laura Arribas-Hernández, Funding acquisition, Investigation, Methodology, Resources, Supervision, Validation, Visualization, Writing – original draft, Writing – review and editing; Sarah Rennie, Conceptualization, Investigation, Methodology, Resources, Software, Supervision, Visualization, Writing – original draft; Tino Köster, Investigation, Methodology, Resources, Supervision, Writing – review and editing; Carlotta Porcelli, Data curation, Formal analysis, Methodology, Visualization; Martin Lewinski, Formal analysis, Methodology, Software, Writing – original draft; Dorothee Staiger, Funding acquisition, Methodology, Resources, Supervision, Writing – review and editing; Robin Andersson, Funding acquisition, Supervision, Writing – review and editing; Peter Brodersen, Conceptualization, Funding acquisition, Project administration, Supervision, Validation, Writing – original draft, Writing – review and editing

### Author ORCIDs

Laura Arribas-Hernández http://orcid.org/0000-0003-0605-0407
Carlotta Porcelli http://orcid.org/0000-0003-4675-4898
Dorothee Staiger http://orcid.org/0000-0002-1341-1381
Robin Andersson http://orcid.org/0000-0003-1516-879X
Peter Brodersen http://orcid.org/0000-0003-1083-1150

### Decision letter and Author response

Decision letter https://doi.org/10.7554/eLife.72375.sa1
Author response https://doi.org/10.7554/eLife.72375.sa2

## Additional files

### Supplementary files
- Supplementary file 1. ECT2 HyperTRIBE data.
- Supplementary file 2. ECT2-mCherry iCLIP data.
- Supplementary file 3. Analysis of HOMER-identified and additional motifs around ECT2-iCLIP and m⁶A sites.
- Transparent reporting form

### Data availability
All sequencing data (iCLIP-seq, HyperTRIBE, mRNA-seq, small RNA-seq) have been deposited in the European Nucleotide Archive under accession code PRJEB44359.

The code specific for this article is available at GitHub https://github.com/sarah-ku/targets_arabidopsis (copy archived at swh:1:rev:ab778b60f735a07d2ef181edc5b2dfbf25153021).

The following dataset was generated:

| Author(s) | Year | Dataset title | Dataset URL | Database and Identifier |
|---|---|---|---|---|
| Brodersen P | 2021 | Principles of mRNA targeting and regulation via Arabidopsis YTHDF proteins | https://www.ebi.ac.uk/ena/browser/view/PRJEB44359 | European Nucleotide Archive, PRJEB44359 |

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

## Appendix 1

| DNA oligonucleotides (all sequences are 5' to 3') |
| --- |

**Cloning**

| Constructs | Primer pairs (fragments) |
| --- | --- |
| *ECT2pro:ECT2-FLAG-ADAR-ECT2ter* (in pCAMBIA3300-U) | LA336-695 (*ECT2pro:ECT2*), LA696-615 (*FLAG-ADAR*), LA616-337 (*ECT2ter*) |
| *ECT2pro:FLAG-ADAR-ECT2ter* (in pCAMBIA3300-U) | LA336-697 (*ECT2pro*), LA698-615 (*FLAG-ADAR*), LA616-337 (*ECT2ter*) |
| *AGO1pro:FLAG-ADAR-AGO1ter* (in pCAMBIA3300-U) | MVUSER1-11 (*AGO1pro-FLAG*), MVUSER12-22 (*ADAR*), MVUSER23-6 (*AGO1ter*) |
| *FLAG-ADAR* (in pGEM-T Easy) | LA696-615 (*FLAG-ADAR*) |

**Primers for USER cloning**

| | |
| --- | --- |
| LA336.U-ECT2P.F | GGCTTAAUAAGCAACGAACCAAGGGAAGACG |
| LA337.ECT2T-U.R | GGTTTAAUAGGTTCTCTCGGCTTCTTTGAC |
| LA615.dADAR/ECT2T.R | AGTTATUCGGCAAGACCGAACTCGTC |
| LA616.dADAR/ECT2T.F | AATAACUAAGAGGATGGTGTCGCTC |
| LA695.ECT2/FLAG.R | ATCGCAACCAUTTGCCACCACATCG |
| LA696.ECT2/FLAG.F | ATGGTTGCGAUTACAAGGATGACGATGAC |
| LA697.ECT2P/FLAG.R | AATCCAUGAGAGGAGATTCGACAAACAAAG |
| LA698.ECT2P/FLAG.F | ATGGATUACAAGGATGACGATGAC |
| MVUSER1.F | GGCTTAAUCTATCCAAATTCCAAACCATACG |
| MVUSER6.R | GGTTTAAUGATTCTGTCGATTGCTTTGCTGG |
| MVUSER11.R | ATTGGACTGUACTTGTCATCGTCATCCTTG |
| MVUSER12.F | ACAGTCCAAUGGTGGTGCCACAG |
| MVUSER22.R | ACTGCGGCAGCUCATTCGGCAAGACCGAACTCG |
| MVUSER23.F | AGCTGCCGCAGUTGATTCACCCTCTATCTATCTTTATGACC |

**Primers for site-directed mutagenesis (QuickChange)**

| | |
| --- | --- |
| LA729.dADAR_E488Q_QC.F | CAAAATCGAGTCCGGTCAGGGGACGATTCCAG |
| LA730.dADAR_E488Q_QC.R | CTGGAATCGTCCCCTGACCGGACTCGATTTTG |

**Primers for detection of point mutations**

| | |
| --- | --- |
| LA660.dADAR_E488Q_CP(NlaIII).F | CGAAAACGACACTGGTGTTG |
| LA735.dADAR_E488Q_CP(NlaIII).R | GCTTTTCACTGGAATCGTCCCAT |

