## [Decision Letter]

**Acceptance summary:**

We found your work of general interest to the readers of *eLife*. It represents an important contribution to the fields of RNA biology and plant molecular biology. Your work help explaining how proteins recognize, bind and regulate methylated RNA in plants. This work also provides valuable datasets to the community. Congratulations on this outstanding work.

**Decision letter after peer review:**

Thank you for submitting your article "Principles of mRNA targeting via the Arabidopsis m^6^A-binding protein ECT2" for consideration by *eLife*. Your article has been reviewed by 2 peer reviewers, including Pablo A Manavella as Reviewing Editor and Reviewer #1, and the evaluation has been overseen Jürgen Kleine-Vehn as the Senior Editor. The following individual involved in review of your submission has agreed to reveal their identity: Rupert Fray (Reviewer #2).

Essential revisions:

Both reviewers agreed that the manuscript is well written and present valuable data. No major concerns or additional experimental requests have arisen. We provide here a series of recommendations to further discuss in the manuscript.

1. The authors could perhaps cite Bodi et al., "Adenosine methylation in Arabidopsis mRNA is associated with the 3 ' end, and reduced levels cause developmental defects" at line 53. This paper precedes the other references (albeit by a few weeks) in showing that m6A is associated with mRNA 3' ends (although by a direct biochemical rather than sequencing approach).

2. Line 331 – do the authors think that this could also be that Nanopore is not yet sufficiently robust to call all m6A sites?

3. It could be helpful for the reader to include a paragraph discussing how extensive their finding could be to other YTH proteins and whether other family members can target the URUAY motif.

4. Please also refer to the recommendations made to the companion manuscript for related comments.

---

## [Author Response]

Essential revisions:Both reviewers agreed that the manuscript is well written and present valuable data. No major concerns or additional experimental requests have arisen. We provide here a series of recommendations to further discuss in the manuscript.1. The authors could perhaps cite Bodi et al., "Adenosine methylation in Arabidopsis mRNA is associated with the 3 ' end, and reduced levels cause developmental defects" at line 53. This paper precedes the other references (albeit by a few weeks) in showing that m6A is associated with mRNA 3' ends (although by a direct biochemical rather than sequencing approach).

We completely agree, and have included the suggested reference at this point in the paper. The reference was indeed omitted by mistake in the originally submitted version, and we thank the reviewers for pointing out this mistake.

2. Line 331 – do the authors think that this could also be that Nanopore is not yet sufficiently robust to call all m6A sites?

Although there is little doubt that present ONT sequencing efforts do not identify all m6A sites, in particular due to limiting sequencing depth, we do not think that m6A sites potentially missed by ONT sequencing explain the systematic shift between m6A-sites called by ONT and crosslink sites to ECT2 detected upon UV illumination. As discussed in the manuscript, the fact that miCLIP sites – found by crosslinking of an m6A-specific antibody to pure mRNA in vitro – largely agree with ONT sites, but overlap with ECT2 crosslinking sites, strongly suggests that the shift arises as a consequence of the bias of RNA-protein crosslinks to occur via uridines that are depleted exactly at m6A sites. For this reason, we have not found it necessary to amend the original manuscript on this specific point.

3. It could be helpful for the reader to include a paragraph discussing how extensive their finding could be to other YTH proteins and whether other family members can target the URUAY motif.

We agree, and have done two improvements in this regard:

(1) We have included a few sentences to highlight the possibility that IDR-specific RNA contacts could lead to YTHDF paralogs having distinct target mRNAs, even if co-expressed in the same cells. Specifically, we have expanded the discussion from line 560 as shown below:

“and is reminiscent of the recent demonstration that transcription factors use their globular DNA-binding domains to recognize core sequence elements of promoters, and their IDRs to provide additional DNA contacts, contributing to specificity (Brodsky et al., 2020). Similarly, it is possible that diverging IDRs among YTHDF paralogs could confer target specificity via binding to distinct motifs in the vicinity of m6A sites, such that specific YTHDF-target mRNA repertoires could exist even for YTHDF proteins co-expressed in the same cells.”

(2) We have raised the specific possibility that an ECT2-competing factor could indeed becanother YTHDF protein using IDR-URUAY contacts to competitive binding. That sectioncof the discussion now reads (line 590):

“URUAY may be a site of competition between the IDR of ECT2 and another, as yet unknown, RNA binding protein. Such a competing factor could in theory be another YTHDF protein using higher-affinity IDR-URUAY contacts than ECT2 to achieve competitive binding. Many other possibilities exist, however. For example, it is intriguing that URUAY resembles part of a Pumilio binding site (Hafner et al., 2010; Huh et al., 2013), as it raises the tantalizing possibility of functional interaction between YTHDF and Pumilio proteins.”

4. Please also refer to the recommendations made to the companion manuscript for related comments.

We have done our very best avoid repetition between the two manuscripts. In particular, we have removed all non-essential information in the Introduction of each paper that is already presented in the other, and eliminated redundancy in the Methods of the companion manuscript referring to the equivalent information in this one (that has been unified as a unique Methods section in the main text to eliminate the original Supplemental Methods file).

Avoiding repetitive content in the Results section has been more problematic though: ECT3 HyperTRIBE results are only shown and analyzed in the companion paper, but since it is an independent data set, it must be subjected to the same rigorous controls as is the ECT2 data set described here. Because the results for both proteins are similar, that inevitably causes a similar structure of those parts of the papers that describes the HyperTRIBE data – we do not see a way around that. Nonetheless, we have shortened the description of ECT3 HyperTRIBE in the companion paper by referring to the corresponding parts of this paper whenever possible.